# SPARe: Stacked Parallelism with Adaptive Reordering for Fault-Tolerant LLM Pretraining Systems with $100\text{k}+$ GPUs

**Jin Lee** [1 2]   **Zhonghao Chen** [3]   **Xuhang He** [3]   **Robert Underwood** [2]   **Bogdan Nicolae** [2]
**Franck Cappello** [2]   **Xiaoyi Lu** [3]   **Sheng Di** [2]   **Zheng Zhang** [1]

## Abstract

In large-scale LLM pretraining systems with $100\text{k}+$ GPUs, failures become the norm rather than the exception, and restart costs can dominate wall-clock training time. However, existing fault-tolerance mechanisms are largely unprepared for this restart-dominant regime. To address this challenge, we propose SPARe—Stacked Parallelism with Adaptive Reordering—a fault-tolerance framework that masks node failures during gradient synchronization by stacking redundant data shards across parallelism groups and adaptively reordering execution. SPARe achieves availability comparable to traditional replication while maintaining near-constant computation overhead of only $2 \sim 3\times$, even under high redundancy where traidional replication would require linearly inflating overhead. We derive closed-form expressions for endurable failure count and computation overhead, validate them via discrete-event simulation, and jointly optimize redundancy and checkpointing to minimize time-to-train. At extreme scale with up to $600\text{k}$ GPUs, SPARe reduces time-to-train by $40 \sim 50\%$ compared to traditional replication.

## 1. Introduction

The industrial high-performance computing (HPC) system has scaled up to over $100\text{k}+$ GPUs to meet the computing demand of large-scale pre-training of AI foundation models. As the number of GPUs ($\#\text{GPU}$) increases, the mean time between failures (MTBF) of system decreases as $\mathcal{O}(\frac{1}{\#\text{GPU}})$ (Kokolis et al., 2025; Ostrouchov et al., 2020). Meanwhile, upon system failures, the restarting latency inflates quadratically or worse by default due to global collectives initialization and synchronization (Jiang et al., 2024; Si et al., 2025). As a result, cumulative waste of system downtime from frequent failures grows rapidly, leading to lower system availability[1] and longer time-to-train[2]. The LlaMa-3 paper (Dubey et al., 2024) (Section 3.3.4) reports that the average failure rate is around $4$ interruptions per $1000$ servers (8 GPUs per server) per day. This corresponds to one failure every three hours on average for a $16\text{k}$ H100-GPU system. Extrapolating using the empirically validated MTBF scaling model in Kokolis et al. (2025), failures would occur every $30$ minutes at $96\text{k}$ GPUs and every five minutes at $600\text{k}$ GPUs. This projection is already harsh, yet still optimistic, as modern pretraining systems are increasingly heterogeneous, which further exacerbates failure rates (Jones, 2020; Jayaram Subramanya et al., 2023).

As failure-induced global restarts now happen once per hour or more, *collective communication (re)initialization* once neglected now dominates runtime. At $96\text{k}$ GPUs, Si et al. (2025) show that $\text{NCCL\_init}$ and similar routines become a major bottleneck; even after optimization (Jiang et al., 2024; Si et al., 2025), key collectives still scale linearly with $\#\text{GPU}$. Hu et al. (2025) similarly report that large messages throughput in NCCL is best with linear-scaling algorithms. As a result, restart latency is projected to grow linearly with $\#\text{GPU}$ while MTBF shrinks inversely, pushing LLM pretraining system towards a *restart-dominant regime* where expected restart latency prevails over useful work time. For example, a $600\text{k}+$ cluster of 5-min MTBF may spend $\geq$ 60-min on global restart after every failure. To address this challenge, three major approaches have been considered to achieve fault-tolerance in distributed training: checkpointing, partial recovery, and replication.

Checkpointing attempts to minimize rework burden from progress loss by failures. Universal checkpointing (Lian et al., 2025) decouples checkpoint state from a fixed par-

[1]Department of ECE, University of California, Santa Barbara, United States [2]Argonne National Laboratory, Illinois, United States [3]Department of ECE, University of Florida, United States. Correspondence to:  Jin Lee <hojin@ucsb.edu>, Sheng Di <sdi1@anl.gov>, Zheng Zhang <zhengzhang@ece.ucsb.edu>.

*Proceedings of the $43^{rd}$ International Conference on Machine Learning*, Seoul, South Korea. PMLR 306, 2026. Copyright 2026 by the author(s).

---

[1]Fraction of time the system is operational: uptime divided by total wall-clock time (Avizienis et al., 2004).

[2]Total wall-clock time of training. Job Completion Time.

allelism configuration to enable flexible and fast recovery. GEMINI (Wang et al., 2023b) maintains lightweight in-memory snapshots for near-instant rollback. Just-in-Time checkpointing (Gupta et al., 2024) adapts snapshot timing to high risk windows. Oobleck (Jang et al., 2023) reduces replay cost in pipeline parallelism (Huang et al., 2019) by replaying short segments. Multi-level checkpointing (Di et al., 2014; Maurya et al., 2024; 2026) amortizes I/O across device, host, and storage tiers. These advances lighten the rework burden *once the system is back on the job*. In the restart-dominant regime, however, the performance bottleneck shifts to restart downtime: the cumulative waste spent *before the system can resume the job*. In this setting, reducing the number of global restarts becomes critical to improving system availability hence reducing time-to-train.

In parallel, traditional replication methods have been revisited to improve availability by masking failures with redundant computation, and partial recovery has been proposed to replace global restarts with cheaper, local system recoveries (Losada et al., 2020). Ferreira et al. (2011) disclosed that even minimal redundancy $r = 2$ can tolerate a large number of failures and Benoit et al. (2019) proposed an optimized strategy to minimize time-to-train with checkpointing. However, replication is constrained by a fundamental ceiling: degree-$r$ replication incurs $r\times$ more computation, which becomes prohibitive in practice. This strongly motivates the need for a direct yet practical approach to improving availability without extravagant waste, especially as we are at the doorstep of the restart-dominant regime.

We hereby propose SPARe: Stacked Parallelism with Adaptive Reordering, a failure-masking scheme as effective as traditional replication yet pays only near-constant overhead around $2 \sim 3\times$ even for high redundancy like $r \sim 20$. Instead of fully replicating computation, SPARe stacks shards of computation across synchronous Data Parallelism (Zinkevich et al., 2010; Sergeev & Del Balso, 2018) so that redundant workload can be minimized by adaptively reordering the stacks. SPARe operates entirely on the data-parallel layer hence agnostic to any model architecture and inner parallelism topology. Our contributions in this work are:

- We derive working formulas for the expected failure count and computation overhead SPARe endures and pays, and validate them with our simulation results.

- We optimize SPARe with checkpointing (SPARe+CKPT) to find optimal redundancy $r$ to minimize time-to-train.

- Lastly, we provide discrete-event simulations using the core components of FedDES (Chen et al., 2025), a large parallel system simulation toolkit built on top of SimGrid (Casanova et al., 2014). With realistic system parameters for a $600k$ H100 cluster, we benchmark SPARe+CKPT against replication+CKPT and CKPT-only, showing $40 \sim 50\%$ gain in time-to-train.

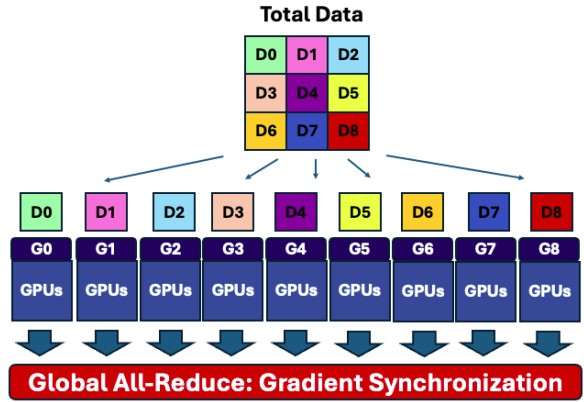

*Figure 1.* Synchronous Data Parallelism.

## 2. Background

### 2.1. Synchronous Data Parallelism

Fig. 1 shows synchronous Data Parallelism (Zinkevich et al., 2010; Sergeev & Del Balso, 2018) under a Megatron-style hybrid-parallel topology (Shoeybi et al., 2019). Let $N$ be the data-parallel degree and $M$ be the size of each model-parallel group. The system consists of $N$ model-parallel groups each containing $M$ GPUs and collectively holding one model replica. Each group may internally combine tensor, pipeline, sequence, or expert parallelism (Narayanan et al., 2021; Huang et al., 2019; Korthikanti et al., 2023; Fedus et al., 2022); in such settings, a single node failure typically interrupts the entire group (Salpekar et al., 2026). Throughout this paper, a *group* refers to one model-parallel group, equivalently one logical data-parallel replica.

This layout induces $M$ data-parallel groups. For each local model-parallel rank $m \in \{0, \ldots, M-1\}$, the same-rank GPUs across the groups form one data-parallel group $\{(0, m), (1, m), \ldots, (N-1, m)\}$, synchronized by a communicator of world size $N$. Thus, the system has $M$ data-parallel communicators in total. Following recent scaling trends (Dubey et al., 2024; Chu et al., 2025), $M$ can be hundreds to a few thousands; we consider $N \sim 10^{2-3}$. See App. F.1 for the communicator topology figure.

At each training step, group $i$ computes a *partial gradient* $\mathbf{g}_i$ from data shard $D_i$. The data-parallel communicators then all-reduce the logical *full gradient* $\overline{\mathbf{g}} = \frac{1}{N} \sum_{i=0}^{N-1} \mathbf{g}_i$ and distribute the corresponding gradient shards for the model update. Hence, all $N$ partial gradients must be collectible; losing any group makes synchronization unavailable and interrupts vanilla Data Parallelism.

### 2.2. Node Failures in Large Parallel HPC System

In large GPU clusters, node failures occur when nodes are forcibly lost by fail-stop faults such as GPU memory error, driver/kernel faults, process crash, power/thermal ex-

cursions and etc., and the aggregate failure rate rises with #GPU (Schroeder & Gibson, 2009). Kokolis et al. (2025) empirically validated that MTBF decreases by $\propto \frac{1}{\#GPU}$ for jobs with more than 32 GPUs. As MTBF decreases, time wasted outside of training increases, hence the total wall-clock time to finish training also increases.

The well-known Young & Daly's formula (Young, 1974; Daly, 2006) gives optimal checkpointing period that minimizes the time to finish training when the rework waste on lost progress is the primary bottleneck. However, in restart-dominant regime, system downtime is of main concern; in this setting minimizing the *portion* of the downtime, hence maximizing the system availability, leads to the minimal time to finish training. For system failure interval $T_f$, checkpointing period $T_c$, checkpoint save time $T_s$, and global restart cost $T_r$, recent work of Saxena et al. (2024) gives optimal checkpointing period for maximal availability as:

$$T_c^\star = T_s + \sqrt{T_s^2 + 2\,T_s\,(T_f + T_r)}, , \qquad (1)$$

where corresponding maximal system availability is:

$$A^\star(T_f, T_s, T_c^\star, T_r) = \frac{T_f - \frac{T_f T_s}{T_c^\star}}{T_f + \frac{T_c^\star}{2} + T_r}. \qquad (2)$$

In our work $T_r$, $T_s$ are fixed parameters decided by model size and system capacity, while $T_f$ is a design factor decided by redundancy $r$ for given data-parallel degree $N$. Hence we denote $T_c^\star$ and $A^\star$ as functions of $T_f$: $T_c^\star(T_f)$, $A^\star(T_f)$.

### 2.3. Partial Recovery: Avoiding Global Restart

Partial recovery minimizes downtime by localizing recovery across surviving nodes so that system can resume job without going through the costly global restart (Bland et al., 2013). For example, **(1) communicator shrinking** excludes failed nodes (nodes) from the job and forms a new communicator over the survivors (Losada et al., 2020; ncc, 2025), **(2) Non-shrinking replacement** respawns failed nodes or activates hot spares to merge them back in (Losada et al., 2020), **(3) Rollback with message logging** restarts only the failed nodes from a checkpoint and uses logged/replayed messages to restore a consistent global state (Elnozahy et al., 2002), **(4) Runtime reinitialization** rebuilds runtime and communicator state through the batch system (Georgakoudis et al., 2020), and **(5) Task migration** dynamically remaps failed work units onto healthy resources (Chakravorty et al., 2006).

In this paper, we consider shrinking paired with replication or checkpointing as the default partial recovery, as it is fastest at re-establishing collective communication across surviving groups and also directly supported by NCCL (ncc, 2025), PyTorch (tor, 2024), and MPI (Losada et al., 2020). Moreover, Bland et al. (2015) report shrinking takes tens of milliseconds for hundreds of nodes. As the data-parallel

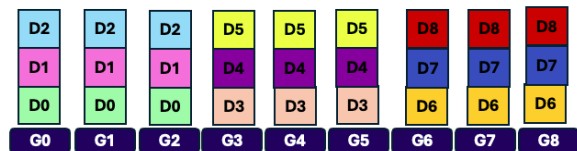

*Figure 2.* Traditional Replication $r = 3$.

communicators are of world size hundreds to a few thousands, we therefore treat their shrink costs as negligible: we presume the job resumes without any meaningful delay after each failure if partial recovery is available for the communicators as long as all partial gradients are collectible.

### 2.4. Traditional Replication: Robust yet Expensive

Fig. 2 shows replication of degree $r = 3$. We define a term, *type*, to denote shard identity: type $i$ is the partial gradient contribution associated with data shard $D_i$. With partial recovery, system can endure multiple failures without job interruption as long as all types of shards are surviving on the job. However, assuming fixed GPU budget, each group now hosts $3\times$ shards compared to the original Data Parallelism, hence $3\times$ workload if we regard the data size as equivalent to the computation amount for simplicity. Assuming random independent failures, higher redundancy $r$ masks more failures before depletion (Ferreira et al., 2011), yet also linearly inflates the workload by $r\times$, a prohibitive overhead that degrades time-to-train in practice.

## 3. SPARe: Stacked Parallelism with Adaptive Reordering

Now we present the SPARe framework. In the imminent restart-dominant regime, one way to improve the system availability (2) to a practical level ($\geq 90\%$) is to increase the system failure interval $T_f$, especially since reducing the global restart cost $T_r$ below the linear scalability has been revealed to be a very subtle problem (Si et al., 2025; Hu et al., 2025). Traditional replication easily increases $T_f$ by masking multiple node failures with redundant computation, yet the linearly inflating computation overhead is prohibitive in practice. SPARe aims to answer to this challenge:

*Can we mask frequent failures with redundant computation while keeping the overhead down to be near-constant?*

Sec. 3.1 introduces the key ideas of SPARe to solve this challenge, and Sec. 3.2 explains the algorithm flow of SPARe.

### 3.1. Key Ideas of SPARe

Traditional replication requires the groups to compute all assigned data to collect all partial gradients. Key intuition behind SPARe is to replicate *shards* of computation (data) instead of fully replicating each group, and stack them across

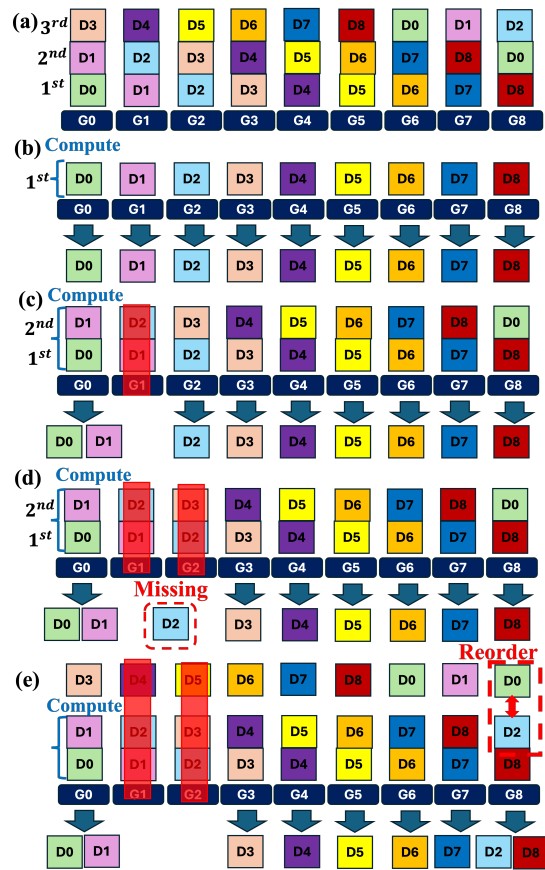

**Algorithm 1** SPARe training loop

**Require:** active groups each with shard stacks
1: all groups active; all-reduce stack← 1
2: **while** training **do**
3:   **for** $j = 1$ **to** all-reduce stack **do**
4:     each active group computes $j^{\text{th}}$ stack
5:   **end for**
6:   all-reduce
7:   **if** success **then**
8:     update parameters; **continue**
9:   **end if**
10:   detect node failure(s) and failed group(s);
11:   RECTLR detects system failure
12:   **if** system failure **then**
13:     global restart; all groups active; reset shard stacks; reset all-reduce stack← 1; **continue**
14:   **end if**
15:   RECTLR decides if reordering is needed
16:   **if** reordering needed **then**
17:     find minimal all-reduce stack; reorder;
18:   **end if**
19:   compute shard(s) lost by failure(s) on current step;
20:   communicator shrink; all-reduce; update parameters;
21:   commit new all-reduce stack and reordered stacks; **continue**
22: **end while**

*Figure 3.* **(a)**: Example of SPARe at $N = 9$, $r = 3$. **(b)**: Before any failure, all partial gradients can be collected after computing the $1^{\text{st}}$ stack. **(c)**: With group 1 failure, system needs to compute up to $2^{\text{nd}}$ stack to collect all types. **(d)**: If group 2 fails later, type 2 partial gradient cannot be collected within the $2^{\text{nd}}$ stack of shards. **(e)**: However, all partial gradients can be collected after computing up to $2^{\text{nd}}$ stack when group 8 stack is reordered.

parallelism so that all types of partial gradients are collectible *before* computing all assigned data (stacks) at each group. Below are the key steps of SPARe of redundancy $r$:

- From synchronous Data Parallelism introduced in Sec. 2.1, replicate all $N$ types of data shards $\{D_0, D_1, \cdots, D_{N-1}\}$ and stack them up to $r$ stacks with cyclic rotation across parallelism so that all types are present in every stack.

- At each training step, schedule gradient synchronization right after all types of shards are computed: aggregate partial gradients as soon as all types are collectible.

- To compute all types of shards within the minimal number of stacks at each training step, reorder the shard stack in each group accordingly to the failure accumulation.

In this way, redundant computation needed to mask failures can be kept down to be minimal at each training step. Also, the adaptive reordering changes only the supplier of each

shard type, not the collected full gradient, hence leaves the optimizer state and the resulting update unchanged. With judicious scheduling of gradient synchronization and failure-adaptive reordering, SPARe needs to compute only $2 \sim 2.8$ stacks in average to collect all partial gradients at each training step even for high redundancies such as $r = 20$, whereas traditional replication requires to finish computing all $r = 20$ stacks amount of data in every training step. Operating entirely at the data-parallel layer, SPARe is agnostic to model architecture and inner parallelism topology.

### 3.2. Algorithm Flow of SPARe

Before we explain the algorithm flow of SPARe we first define the failure ontology we use throughout this work. We denote an individual GPU failure as a *node failure*. In Data Parallelism any node failure interrupts the job and enforces global restart. However, for replication and SPARe, global restart occurs only when any shard type $i$ depletes with failure accumulation: we refer to this incident as the *wipe-out* of shard $i$, and also as *system failure*. Lastly, we assume the node failures are detected when the system calls all-reduce across data-parallel groups for gradient synchronization, following the typical large parallel system convention.

The training loop of SPARe develops on the synchronous Data Parallelism introduced in Sec. 2.1, except that it does

---

**Algorithm 2** RECTLR algorithm

---

**Require:** current shard stacks, current all-reduce stack
**Ensure:** Either RESTART or updated stacks
 1: $S_0 \leftarrow$ all-reduce stack
 2: **Phase 0: decide if reordering is needed.**
 3: **if** HK-FIXED succeeds **then**
 4:     **return** current stacks {no reordering}
 5: **end if**
 6: **Phase 1: find minimal all-reduce stack.**
 7: $S^\star \leftarrow$ UNDEFINED
 8: **for** $S = S_0$ **to** $r$ **do**
 9:     **if** HK-FREE succeeds **then**
10:         $S^\star \leftarrow S$; **break**
11:     **end if**
12: **end for**
13: **if** $S^\star$ is UNDEFINED **then**
14:     flag system failure to trigger global restart
15: **end if**
16: all-reduce stack$\leftarrow S^\star$
17: **Phase 2: reorder with minimal movement.**
18: run MCMF on current stacks and all-reduce stack
19: **return** updated all-reduce stack and reordered stacks

---

not compute all the stacks to complete a training step. Instead, it triggers global all-reduce as soon as a committed number of stacks are computed, which is aimed to be the minimal stacks required to collect all partial gradients so that the redundant computation can be minimized at each step. We denote this committed number as *all-reduce stack*, which is set as 1 by default when training starts.

When node failure happens, the next all-reduce will fail with collective hang/drop and system will detect the failure. SPARe then initiates the *reordering controller*, RECTLR, which serves three purposes:

- System failure detection;

- Finding the minimal all-reduce stack for subsequent steps;

- Reorder stacks accordingly with minimal movement.

If RECTLR detects system failure, system undergoes global restart, reset the stacks to the original order and the all-reduce stack back to be 1. If not, RECTLR finds the minimal all-reduce stack with the new failure(s) and reorder the stacks accordingly. After RECTLR, system needs to collect the missing partial gradients lost by the new failure(s) to complete the current training step, hence commands all surviving model-parallel groups hosting the missing shard types to compute them. We denote this additional computation stack as *patch compute*. After patch compute, the system shrinks the communicators across all

data-parallel groups, performs gradient synchronization and model updates, commits the new all-reduce stack and reordered stacks, then proceeds to the next training step. See Alg. (1) for the pseudo-code of the SPARe training loop.

RECTLR plays the central role in SPARe training loop and it runs with 3 phases:

- **Phase 0** decides if reordering is needed with the new failure(s), by checking if the current all-reduce stack can collect all partial gradients across data-parallel groups via Hopcroft-Karp (HK) algorithm (Hopcroft & Karp, 1973) on the *fixed* stacks of shards; we denote it as HK-FIXED. HK algorithm checks the bipartite graph feasibility between $N$ shard types to the stacks of surviving model-parallel groups up to the all-reduce stack. If HK-FIXED succeeds, no reordering is needed, hence RECTLR quits.

- **Phase 1** searches for the minimal all-reduce stack that can collect all partial gradients, by incrementally iterating the HK algorithm from the previous all-reduce stack up to redundancy degree $r$, now allowing *free permutation* of shard stacks within each group (HK-FREE). If no HK-FREE iteration succeeds, a wipe-out has occurred, and RECTLR flags a system failure to trigger a global restart.

- **Phase 2** finds how to reorder the stacks with minimal movement to achieve the newly found minimal all-reduce stack, by running min-cost max-flow algorithm (Goldberg & Tarjan, 1990), MCMF.

See Alg. (2) for the pseudo-code of RECTLR. At $N \sim 10^{2-3}$, HK-FIXED, HK-FREE, and MCMF do not require significant computational cost. See App. D for the detailed descriptions and complexity analysis of RECTLR.

## 4. Theoretical Analysis of SPARe

In this Section, we provide theoretical analysis on three key properties of SPARe:

- How many failures can SPARe endure before wipe-out?

- How much computation overhead SPARe needs to pay?

- What is the optimal redundancy for minimal time-to-train when merged with checkpointing?

We provide key results on the first two questions in Sec. 4.1 and show joint optimization with checkpointing in Sec. 4.2. In our theoretical analysis, we assume exponential distribution on node failures (uniform interval). However, we consider the realistic Weibull distribution (Weibull, 1951; Schroeder & Gibson, 2009; Wang et al., 2023a; Min et al., 2025) in our simulations in Sec. 5.

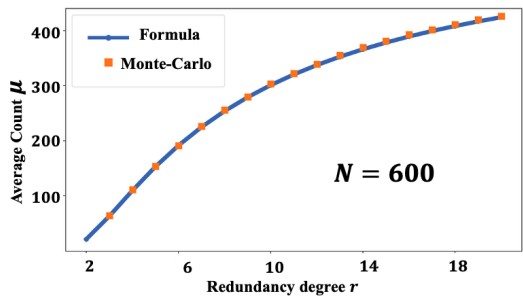

*Figure 4.* Average endurable failure count by redundancy $r$.

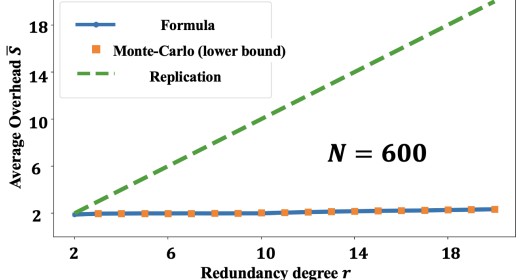

*Figure 5.* Average computation overhead by redundancy $r$.

### 4.1. Theoretical Results about SPARe

In order to minimize the impact of each individual node failure, we design SPARe so that no pair of different shard types overlap in more than one group as in Fig. 3. See App. B.1 for the shard distribution rule that achieves this goal. In this way, wipe-out incidents of different shard types are almost independent to each other, hence their statistics can be approximated to follow the Poisson distribution (Chen, 1975; Barbour & Eagleson, 1983). This leads to two key results:

- SPARe can asymptotically endure as many failures as traditional replication: Thm. 4.1.

- SPARe can achieve computation overhead close to its theoretical lower bound: Thm. 4.2.

**Theorem 4.1** (**Average Failure Count**). *The average failure count $\mu(N, r)$ SPARe can mask before the first wipe-out is asymptotically:*

$$\mu(N, r) \approx \boxed{\frac{\Gamma(1/r)}{r} N^{1-1/r}}, \qquad (3)$$

*where $\Gamma$ refers to the Gamma function in NIST (2025).*

*Proof.* According to Barbour & Eagleson (1983), wipe-out incidents are almost independent to each other so that Poisson approximation is valid (Chen, 1975):

$$\mu(N, r) \approx \sum_{k \geq 0} \exp\left(-N\left(\frac{k}{N}\right)^r\right) \approx N \int_0^\infty e^{-Nt^r} dt$$

$$= \frac{1}{r} N^{1-1/r} \Gamma\left(\frac{1}{r}\right) = \frac{\Gamma(1/r)}{r} N^{1-1/r}. \qquad (4)$$

See App. B.2 for the full proof. □

$\mu(N, r)$ derived in (3) is consistent to the traditional replication formula given by (Ferreira et al., 2011). Fig. 4 shows that SPARe can endure up to 426 failures in average for $N = 600$ with redundancy $r = 20$. This means that for a system of MTBF $m$, SPARe effectively increases it up to $426 \times m$, substantially improving the system availability. Traditional replication requires $20\times$ computation overhead for this availability gain. However, with adaptive reordering, SPARe only pays $2.8\times$ for the same availability gain.

**Theorem 4.2** (**Average Computation Overhead**). *The average computation overhead $\overline{S}(N, r)$ SPARe needs to pay before the first wipe-out is approximately*

$$\overline{S}(N, r) \approx \boxed{\frac{1}{\lfloor \mu \rfloor} \sum_{k=0}^{\lfloor \mu \rfloor - 1} \left(c(k) + \rho_k\right)}, \qquad (5)$$

*where $\mu$ is the average failure count of Thm. (4.1) and $c(k)$ is the lower-bound of all-reduce stack: $c(k) := \left\lceil \frac{N}{N-k} \right\rceil$. $\rho_k$ is the probability of patch compute at $k$ failures, $\rho_k := \max\{0, 2N - n_k\}/n_k$, where $n_k := c(k)(N - k)$.*

$\overline{S}(N, r)$ *has an idealistic lower bound of:*

$$\overline{S}(N, r) \gtrsim \frac{1}{\lfloor \mu \rfloor} \sum_{k=0}^{\lfloor \mu \rfloor - 1} c(k), \qquad (6)$$

*achievable if system detects failures earlier than the global all-reduce hence does not require patch computes.*

*Proof.* Probabilities of adversarial cases where all-reduce stack is forced to be $> c(k)$ can be formulated by Hall's Marriage Theorem (Hall, 1987), which are negligible at $N \sim 10^{2-3}$. Therefore all-reduce stack is $\approx c(k)$. Expected number of patch compute per step can be approximated by noting that the probability of a random failure hitting any singleton type[3] scales as $\rho_k$: $\Pr(\text{patch compute} \mid k) \approx \rho_k$. See App. B.3 for the full proof. □

Fig. 5 clearly shows that the average computation overhead of SPARe is near-constant around $2 \sim 2.8\times$ compared to the linearly scaling $r\times$ of traditional replication. As SPARe achieves as high availability as replication, near-constant overhead results in significant gain in time-to-train.

See App. C for Monte-Carlo validation on $\mu(N, r)$ and the lower bound of $\overline{S}(N, r)$, which shows $1.13\%$ and $0.60\%$ absolute error respectively.

**Failure heterogeneity and correlations** Real systems may exhibit heterogeneous failure rates across GPUs and correlated failures within the same allocation group or failure domain (Thorpe et al., 2023; Wan et al., 2025). These

---

[3]Type that is computed only once before all-reduce, hence its loss prevents gradient synchronization.

effects do not change the feasibility logic of SPARe: Alg. (1) and Alg. (2) operate only on the realized failure count and survivor set, and therefore do not require a specific temporal failure law. They may, however, affect the distribution of survivor sets and hence the accuracy of the closed-form averages in Thm. 4.1 and Thm. 4.2. In practice, this can be mitigated by making the shard placement independent of physical failure domains, such as spreading replicas across racks, zones, or parallelism groups, as commonly done in failure-aware training systems such as Bamboo (Thorpe et al., 2023) and ByteRobust (Wan et al., 2025).

## 4.2. Joint Optimization with Checkpointing

SPARe cannot mask failures indefinitely hence must be paired with checkpointing to ensure forward progress under continual system failures. Merging two schemes, SPARe+CKPT, entails two coupled trade-offs: one that hinges on checkpointing period, checkpointing overhead versus rework amount, and the other on redundancy $r$, availability gains versus computation overhead. In the following, we provide the joint optimization of SPARe+CKPT to find the optimal redundancy $r$ and corresponding checkpointing period that minimize time-to-train.

Let $T_0$ be the time-to-train of $N$-way Data Parallelism in no failure scenario. Then the useful work time a SPARe needs to finish is $T_0 \times \overline{S}(N, r)$. Since system availability (2) is the fraction of useful work time over time-to-train, we can set a normalized time-to-train function $J(r)$ as:

$$ J(r) := \frac{\text{time-to-train}}{T_0} = \frac{\overline{S}(N, r)}{A^\star(\mu(N, r)m)}. \qquad (7) $$

where $m$ is the system MTBF on node failures.

**Theorem 4.3** (**Optimal $r^\star$ for minimal time-to-train**). *Using checkpointing period of Eq. (1), SPARe+CKPT achieves minimal time-to-train at optimal redundancy $r^\star$:*

$$ r^\star \approx \boxed{\lfloor \log_2 N + 0.833 \rfloor}. \qquad (8) $$

*Proof.* Substitute Eq. (1) (2) (3) (5) to $J(r)$. $J(r)$ is minimized around $r^\star$ that satisfies $\mu(N, r^\star) \approx N/2$ and $\overline{S}(N, r^\star) \approx 2$. Let $\varepsilon := 1/r^\star$ and use $\Gamma(\varepsilon) = \frac{1}{\varepsilon} - \gamma + O(\varepsilon)$. Solving it with logarithm yields

$$ r^\star \approx \log_2 N + \frac{\gamma}{\ln 2} \approx \log_2 N + 0.833. \qquad (9) $$

See App. B.4 for the full proof. $\square$

## 5. System Performance Evaluation

This section evaluates the performance of SPARe using a discrete-event simulator implemented upon FedDES (Chen et al., 2025), a simulation toolkit developed on top of Sim-Grid (Casanova et al., 2013). SimGrid provides a mature and

*Table 1.* DES system parameters for 600k H100 cluster.

| PARAMETER | SETTING |
| --- | --- |
| FAILURE | MTBF 300s, $k = 0.78$ (WEIBULL) |
| GLOBAL RESTART | $T_r = 3600$s |
| MODEL SIZE | 10T PARAMS (20TB) |
| DP GROUPS | $N \in \{200, 600, 1000\}$ |
| DATA/STACK | 256M TOKENS ($4 \times 64$M) |
| COMPUTE/STACK | $T_{\text{comp}} = 64$s PER STACK |
| TRAIN HORIZON | $T_0 = 10{,}000$ STEPS $\times (T_{\text{comp}} + T_a)$ |
| ALL-REDUCE | $T_a = 2, 6, 10$s AT EACH $N$ |
| FAILED ALL-REDUCE | 50% OF ALL-REDUCE, $0.5 \times T_a$ |
| COMM. SHRINK | 0.1s |
| CKPT TIME | $T_s = 60$s |
| EVENT JITTER | $\times \mathcal{N}(1, 0.05^2)$ ON ALL EVENTS |

validated simulation framework for modeling distributed systems and applications (Binkert et al., 2011; Varga, 2001; Casanova et al., 2008). In discrete-event simulation, system execution is represented as a chronology of time-stamped events, where time advances by repeatedly processing the next scheduled event rather than by stepping through a fixed time grid (Tocher & Owen, 2008; Neuwirth & Paul, 2021). This event-driven abstraction enables controlled and repeatable failure injection and realistic end-to-end training time accounting, even at system scales that are impractical to reproduce experimentally such as a 600k-H100 cluster (Casanova et al., 2014). For further details on our choice of simulator, see App. E. Our simulator compares SPARe+CKPT against two baselines: replication with checkpointing (Rep+CKPT) and standard data parallel training with checkpointing only (CKPT-only), under realistic large-scale system parameters.

### 5.1. Realistic System Parameters

Table 1 shows system parameters we have set to emulate a restart-dominant system of 600k H100s, where MTBF is projected to be $m = 5$-min according to the scaling law validated by Kokolis et al. (2025) and MTBF reported for 16k H100s (Dubey et al., 2024). We set the global restart to take $T_r = 60$-min to test SPARe and baselines on a harsh restart-dominant regime. We set the LLM to be of 10T parameters of memory size 20 TB (FP16). We consider three cases of data-parallel degree: (i) $N = 200$, (ii) $N = 600$, and (iii) $N = 1000$ parallel groups. From the Llama 3 training report of Dubey et al. (2024) and Chu et al. (2025), we project the compute power per GPU to be 400 TFLOPs and one shard of training data to be of size $4 \times 64$M Tokens, considering 4 gradient accumulations. Consequently, we set the compute time per shard to be $T_{\text{comp}} = 64$s[4]. We model the global all-reduce for gradient synchronization to be ring-based (Patarasuk & Yuan, 2009), hence scales up linearly by $N$ and take $T_a = 2, 6, 10$s for each $N$ considering the partial gradient size of 20TB and conservative collective

---

[4]Regardless of $N$, per GPU workload is the same $\frac{256\text{M Tokens}}{400\text{TFLOPs}}$.

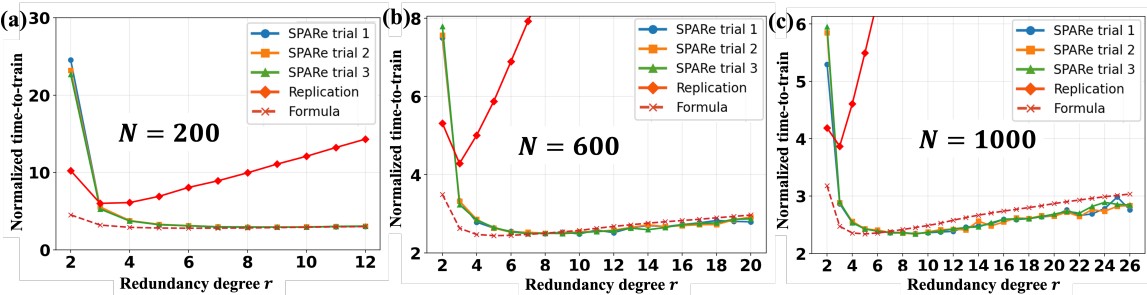

*Figure 6.* Time-to-train/$T_0$ of SPARe+CKPT / Rep+CKPT from simulation and $J(r)$ (7) for **(a)** $N = 200$, **(b)** $N = 600$, **(c)** $N = 1000$.

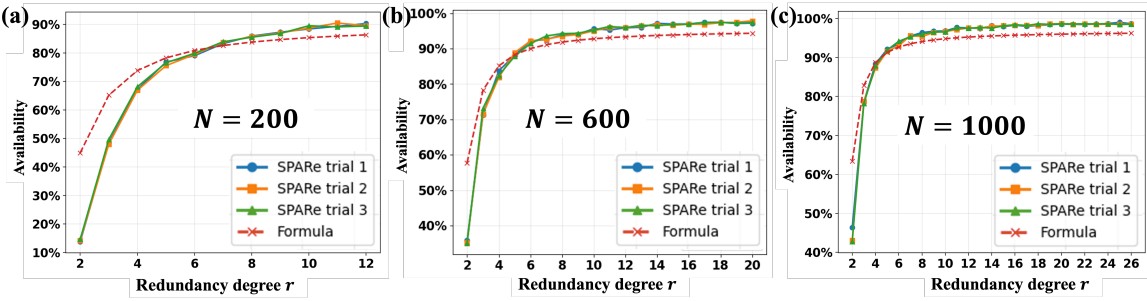

*Figure 7.* Availability of SPARe+CKPT from simulation and $A^\star(\mu m)$ (2) for **(a)** $N = 200$, **(b)** $N = 600$, **(c)** $N = 1000$

goodput per GPU of 400 Gb/s. Therefore, we set a training step that computes one data shard to take $(T_{\text{comp}} + T_a)$ for simplicity, and set 10,000 steps to finish the training: the base time-to-train in the no-failure scenario is $T_0 = 10,000 \times (T_{\text{comp}} + T_a)$.

We inject failure events following the realistic Weibull distribution (Weibull, 1951) and choose the seminal shape parameter of $k = 0.78$ as provided by Schroeder & Gibson (2009). Weibull distribution is widely recognized as a legitimate standard proxy to represent real-life failures at the group-level abstraction as shown in Wang et al. (2023a) and the Titan trace study (Min et al., 2025). As we model a system that detects failed group(s) only at the global all-reduce step, we assume the *failed* all-reduce to take, in expectation, half the time of a successful one, without loss of generality.

Lastly, we set both shrink cost and reordering controller cost to be 0.1s. For the data-parallel degree of $N$, each data-parallel group consists of $N$ GPUs hence the synchronizing communicator is of world size $N$. Therefore, our shrink cost setting is justified by the empirical study of Bland et al. (2015) for given $N = 200, 600, 1000$. Furthermore, checkpoint save time is set to be 1-min and optimal checkpointing period $T_c^\star$ is calculated from Eq. (1). Lastly, compute jitter of normal distribution $\times \mathcal{N}(1, 0.05^2)$ is added to all events to reflect real-life variances and system noise.

### 5.2. Performance Evaluation Results

We created three event trails and ran each for three schemes of SPARe+CKPT, Rep+CKPT, and CKPT-only.

See App. F.2 for the flowchart of each scheme.

#### 5.2.1. RESULTS OF CONVENTIONAL BASELINES

Under the harsh restart-dominant setting, CKPT-only did not proceed more than a few steps in a time where other schemes would finish the training. On the other hand, Rep+CKPT achieved minimal time-to-train at low redundancy $r = 3$ in all cases as shown in Fig. 6, consistent to prior works of Ferreira et al. (2011) and Elliott et al. (2012). For higher redundancies however, Fig. 6 clearly shows that the time-to-train of Rep+CKPT inflates linearly from the $r\times$ overhead.

#### 5.2.2. SPARE RESULTS AND GAIN ANALYSIS

SPARe+CKPT proves to fully exploit the significant availability ($> 90\%$) at high redundancies by keeping the computation overhead low ($2 \sim 3\times$) with failure-adaptive reordering . Fig. 6 shows the normalized time-to-train $J(r) = \text{time-to-train}/T_0$ along with its theoretical prediction from $J(r)$ (7), and Fig. 7 shows empirical availability with theoretical projection $A^\star(\mu m)$ (2). SPARe+CKPT performs better than predicted at high redundancies, showing higher availability and clearly lower time-to-triain. That is because failure rate effectively decreases for high accumulation, as it is proportional to the number of active GPUs (Schroeder & Gibson, 2009; Kokolis et al., 2025); as high $r \to$ high $\mu(N, r)$, hence for higher $r$ and $N$ failure masking scheme is more advantageous. Fig. 8 shows the average number of stacks computed per training step, in other words, the empirical computation overhead along

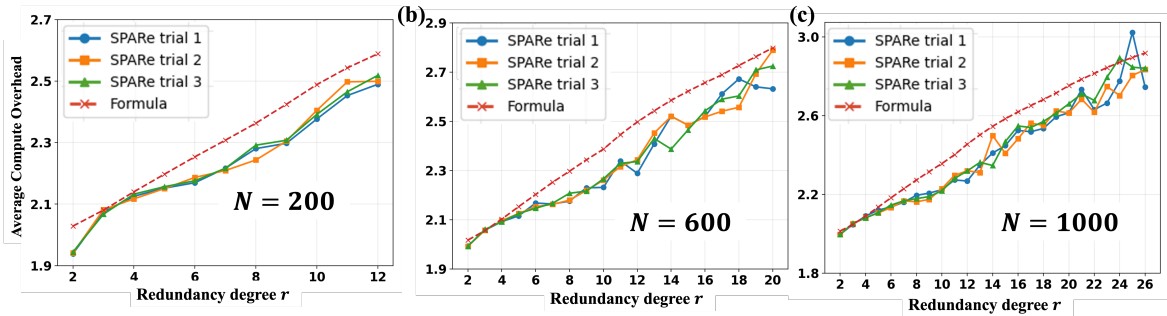

*Figure 8.* Average Computation Overhead of SPARe from simulation and $\overline{S}(N,r)$ (5) for **(a)** $N = 200$, **(b)** $N = 600$, **(c)** $N = 1000$

with its theoretical prediction of $\overline{S}(N,r)$ (5). Our formula closely fits the empirical values within $\leq 4\%$ absolute error, validating Thm. (4.2).

However, SPARe+CKPT performs worse than expected at low redundancies, even worse than Rep+CKPT at $r = 2$. Fig. 8 explicitly shows that the theoretical prediction $\overline{S}$ (5) still closely fits the simulations at low redundancies, indicating our theories on computation overhead are solid. Therefore, the deterioration comes from the $k < 1$ Weibull distribution, where failure accumulates faster at early accumulation: low $r \rightarrow$ low $\mu(N,r)$, hence system encounters the disruptive global restarts much more frequently than our predictions made upon exponential distribution. This implies that this issue does not set a fundamental limit on SPARe, and can be easily improved by employing dynamic-based checkpointing (Bougeret et al., 2011; Benoit et al., 2022) effective for Weibull distribution. We

*Table 2.* Minimum time-to-train averaged over 3 trials.

| $N$ | Rep+CKPT | | SPARe+CKPT | | | Gain |
|---|---|---|---|---|---|---|
| | time-to-train/$T_0$ | Availability | time-to-train/$T_0$ | $r^\star$ | Availability | [%] |
| 200 | 6.07 | 61.74% | 2.92 | 9 | 87.00% | 51.9% |
| 600 | 4.27 | 79.89% | 2.49 | 8 | 93.90% | 41.7% |
| 1000 | 3.88 | 84.41% | 2.34 | 9 | 96.54% | 39.6% |

evaluate the gain of SPARe+CKPT versus Rep+CKPT on the minimal time-to-train/$T_0$ achieved in each scheme and $N$ (Table 2). Compared to the best of Rep+CKPT, the best of SPARe+CKPT achieved $40 \sim 50\%$ gain in time-to-train. Note that the theoretical predictions of the optimal $r^\star$ (8) for SPARe+CKPT are $r^\star = 8, 10, 10$ at $N = 200, 600, 1000$ in Thm. (4.3), showing slight discrepancies coming from the exponential distribution assumption: for low $N$ use SPARe with higher redundancy than $r^\star$ (8), and for high $N$ use lower redundancy than $r^\star$ (8).

## 6. Related Works

Recent fault-tolerance systems for large-scale training reduce recovery cost after failures through storage hierarchy, hot spares, or topology-aware reconfiguration. Storage-assisted checkpointing systems, such as GEMINI (Wang et al., 2023b) and DataStates-LLM (Maurya et al., 2024; 2026), keep recoverable training state in memory or storage tiers to reduce rollback and checkpoint I/O cost; SPARe is orthogonal and complementary, as it reduces the *frequency* of rollback recovery rather than the cost of executing it. Hot-spare and migration methods, such as TrainMover (Lao et al., 2024), reduce interruption time by preparing standby resources and migrating or replacing failed workers; in contrast, SPARe does not replace failed groups during tolerable failures, but shrinks the data-parallel communicators and lets surviving groups take over missing workloads by computing pre-stacked redundant data shards. Topology-aware recovery systems, including Bamboo (Thorpe et al., 2023), ReCycle (Gandhi et al., 2024), and FT-HSDP (Salpekar et al., 2026), exploit specific pipeline, data, or hybrid-parallel layouts to reroute work, adapt schedules, or recover failed training replicas. SPARe operates at a different abstraction layer: it sits above the inner model-parallel topology and only requires every partial-gradient type to remain collectible across surviving model-parallel groups. Thus, these systems can serve as inner-topology recovery substrates, while SPARe masks accumulated group failures without per-failure global restart or full-stack rescheduling.

## 7. Conclusion

In this work, we have introduced SPARe: Stacked Parallelism with Adaptive Reordering, which masks failures as many as traditional replication yet keeps the computation overhead as low as $2 \sim 3\times$ even for high redundancies. SPARe achieves $40 \sim 50\%$ lower time-to-train compared to traditional replication baseline according to our realistic discrete-event simulations, at the harsh restart-dominant system setting projected for 600k H100 GPUs. In the imminent restart-dominant regime, masking failures, hence bypassing global restarts, is the most direct and intuitive strategy. SPARe positions itself as an effective and practical proposal for fault-tolerant LLM pretraining systems with 100k+ GPUs, highly versatile as it is agnostic to any model architecture and inner parallelism topology.

## Impact Statement

This work aims to advance fault-tolerant distributed training system on to the imminent restart-dominant regime, with the goal of achieving system availability over $90\%$ yet keeping computation overhead low to finish the training in practical time span. By substantially reducing the time-to-train required for next generation LLM pretraining, the proposed method of SPARe lowers the cost of frontier foundation model training and further elevate the ceiling for LLM scaling. More broadly, finishing foundation model training faster can contribute to lower energy consumption and more sustainable use of extreme scale HPC clusters. At the same time, increased accessibility to scale up LLM pretraining may accelerate progress of foundation model studies and development, impacting wide range of science and engineering. All experiments in this paper are conducted using publicly available libraries. The method itself is a fault-tolerance protocol for large parallel systems and does not introduce new capabilities that are inherently harmful. We do not foresee any immediate negative societal impacts directly resulting from this work.

## Reproducibility Statement

Codes required to reproduce the experiments are provided in https://github.com/padsysl/SPARe.git, along with the simulation trajectories used in this work. See the flowcharts of each scheme in App. F.2.

## Acknowledgements

We thank Ziyue Liu and Zhengyang Wang for helpful early discussions. This work was supported by the U.S. Department of Energy, Office of Science, Advanced Scientific Computing Research (ASCR), under contract DE-AC02-06CH11357, DE-SC0024207, and DE-SC0025390. This work was also supported by the National Science Foundation under contract CCF-2107321 and OAC-2623546.

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

# A. Notation

*Table 3.* Notation Table.

| SYMBOL | MEANING |
|---|---|
| $N$ | **NUMBER OF DP GROUPS** |
| $r$ | REDUNDANCY DEGREE |
| $G_r^N$ | **OPTIMAL GOLOMB-RULER**: LENGTH $r$ MODULO $N$ |
| $H_i \subset [N]$ | **HOST SET**: INDICES OF GROUPS THAT HOST SHARD TYPE $i$. |
| $T_w \subset [N]$ | **TYPE SET**: INDICES OF SHARD TYPES HOSTED BY GROUP $w$ (SO $|T_w| = r$). |
| $U_k \subset [N]$ | **SURVIVOR SET**: INDICES OF ACTIVE GROUPS AFTER $k$ FAILURES; $|U_k| = N - k$. |
| $\mathrm{STK}[w]$ | **PERSISTENT LOCAL STACK ORDER** AT GROUP $w$: A PERMUTATION OF $T_w$ USED AS ITS COMPUTE SEQUENCE. |
| $\mathrm{STK}[w][j]$ | SHARD TYPE COMPUTED AT THE $j^{\mathrm{th}}$ STACK OF GROUP $w$. |
| $S_A$ | **ALL-REDUCE STACK**: SYSTEM TRIGGERS ALL-REDUCE AFTER ALL ACTIVE GROUPS FINISH $S_A$ STACKS. |
| $S(U_k)$ | **MINIMAL FEASIBLE STACK** SO THAT ALL SHARD TYPES ARE COLLECTIBLE ACROSS SURVIVORS $U_k$. |
| $c(k)$ | **CAPACITY LOWER BOUND** FOR $S(U_k)$: $c(k) := \left\lceil \dfrac{N}{N-k} \right\rceil$, HENCE $S(U_k) \geq c(k)$. |
| $F$ | **FAILURE COUNT TO FIRST WIPE-OUT**: $F := \min\{k \geq r : \exists i \in [N] \text{ S.T. } H_i \cap U_k = \emptyset\}$. |
| $P_k$ | **NO WIPE-OUT PROBABILITY** AT FAILURE COUNT $k$: $P_k := \Pr\{F > k\}$. |
| $\mu(N, r)$ | **AVERAGE ENDURABLE FAILURE COUNT**: $\mu(N, r) := \mathbb{E}[F]$. |
| $q$ | **PATCH-COMPUTE COUNT** PER STEP. |
| $T_f$ | **MEAN TIME BETWEEN SYSTEM FAILURE (GLOBAL RESTART)** |
| $T_s$ | **CKPT SAVE TIME** |
| $T_c$ | **CKPT PERIOD** |
| $T_r$ | **GLOBAL RESTART LATENCY** |

# B. Proofs

## B.1. Mathematical definition of SPARe

**Definition B.1 (Cyclic Golomb Ruler distribution rule).**
Set $G_r^N = \{g_0, \ldots, g_{r-1}\} \subset \mathbb{Z}_N$[5] as the optimal Golomb Ruler (Golomb, 2017) of length $r$, the set of unique pairwise differences modulo $N$ with $g_0 = 0$ and minimal $g_{r-1}$. In other words, the pairwise difference set $\{\pm(g_a - g_b) \bmod N : 0 \leq a < b \leq r - 1\}$ has $r(r-1)$ distinct non-zero elements. A SPARe scheme $(N, r)$ distributes shards of type $i \in [N]$ across $N$ groups with redundancy $r$ as:

$$H_i := \{ (i - g) \bmod N : g \in G_r^N \}, \qquad (10)$$

where $H_i$ is the *host set*, set of group indices hosting shard $i$, with the caveat of $N \geq (2g_{r-1} - 1)$. Equivalently, each

---

[5] $\mathbb{Z}_N := \mathbb{Z}/N\mathbb{Z}$, the additive cyclic group of integers modulo $N$, i.e., residues $\{0, 1, \ldots, N-1\}$ with addition mod $N$

group $w$ hosts *type set*, set of shard indices group $w$ hosts:

$$T_w := \{ (w + g) \bmod N : g \in G_r^N \}. \qquad (11)$$

We provide a lemma to prove that the cyclic Golomb Ruler distribution rule of Def. (B.1) achieves our intention.

**Lemma B.2.** *Following the cyclic Golomb Ruler distribution rule of Def. (B.1), any two distinct shard types of $(N, r)$ share at most one host: $|H_i \cap H_j| \leq 1 (i \neq j)$.*

*Proof.* Assume $i \neq j$ share two hosts $w_1, w_2$ ($|H_i \cap H_j| \geq 2$). Then there would be distinct $g_a, g_b, g_c, g_d \in G_r^N$ which suffice $w_1 \equiv i - g_a \equiv j - g_b$ and $w_2 \equiv i - g_c \equiv j - g_d$ (mod $N$) from Eq. (10). That results as $g_a - g_b \equiv g_c - g_d$ (mod $N$) which contradicts to definition of $G_r^N$. □

## B.2. Proof of Theorem 4.1

*Proof.* Let $F$ be the failure count at first wipe-out and $P_k := \Pr\{F > k\}$. Expanding $\Pr\{F = k\} = P_{k-1} - P_k$, expectation value of $F$ can be written as:

$$\mu(N, r) := \mathbb{E}[F] = \sum_{k=0}^{N-1} P_k. \qquad (12)$$

Fix failure count $k \in \{0, 1, \ldots, N\}$. Conditional on exactly $k$ node failures, the failed set $B_k := U_k^c \subset [N]$ is a uniformly random $k$-subset of $[N]$. For each group $w \in [N]$, define the exchangeable trials

$$Y_w := \mathbf{1}\{w \in B_k\} \in \{0, 1\}. \qquad (13)$$

For each shard type $i \in [N]$, define the wipe-out event and indicator

$$A_i^{(k)} := \{H_i \subseteq B_k\}, \qquad X_i^{(k)} := \mathbf{1}\{A_i^{(k)}\}. \qquad (14)$$

Let the number of wiped-out types after $k$ failures as:

$$W_k := \sum_{i \in [N]} X_i^{(k)} \qquad (15)$$

be the number of wiped-out types after $k$ failures. Then

$$P_k = \Pr\{F > k\} = \Pr\{W_k = 0\}. \qquad (16)$$

Now, we use Barbour & Eagleson (1983, Thm. 1) on $W_k$ to show wipe-out of different shard types are almost independent to each other, hence the statistics of $W_k$ closely follows the Poisson approximation from Chen (1975): following the notations from Barbour & Eagleson (1983), the trials are $\{Y_w\}_{w \in [N]}$ (exchangeable), the subset size is $r$, and the index family is $\mathcal{N} := \{H_i : i \in [N]\}$ (so $|\mathcal{N}| = N$).

Wipe-out probability of type $i$ at $k$ failures can be written as:

$$p_k := \mathbb{E}[X_i^{(k)}] = \Pr(H_i \subseteq B_k) = \frac{\binom{N-r}{k-r}}{\binom{N}{k}} = \frac{(k)_r}{(N)_r}, \tag{17}$$

where $(x)_r := x(x-1)\cdots(x-r+1)$. Hence the expectation value of of the number of types wiped-out at $k$ failures is:

$$\lambda_k := \mathbb{E}[W_k] = Np_k. \tag{18}$$

For $i \neq j$, write $t_{ij} := |H_i \cap H_j|$. By Lem. (B.2), $t_{ij} \leq 1$. Thus only $t = 0$ and $t = 1$ occur, and the corresponding pairwise joint wipe-out probabilities are

$$q_1(k) := \Pr(X_i^{(k)} X_j^{(k)} = 1 \mid t_{ij} = 1) \tag{19}$$
$$= \Pr(|H_i \cup H_j| = 2r - 1 \text{ all fail}) \tag{20}$$
$$= \frac{(k)_{2r-1}}{(N)_{2r-1}}, \tag{21}$$

$$q_0(k) := \Pr(X_i^{(k)} X_j^{(k)} = 1 \mid t_{ij} = 0) = \frac{(k)_{2r}}{(N)_{2r}}, \tag{22}$$

where $q_1$ is joint wipe-out probability between overlapping types and $q_0$ is that of between non-overlapping types.

Let $C_t$ be the number of ordered pairs $(i, j)$ with $|H_i \cap H_j| = t$. Because each group hosts exactly $r$ types (Def. (B.1)) and Lem. (B.2) precludes double-counting across multiple shared hosts,

$$C_1 = N\,r(r-1), \qquad C_t = 0 \text{ for all } t \geq 2. \tag{23}$$

Moreover, for any $J \subset [N]$ with $|J| = r$, the number $i_J$ of host sets intersecting $J$ satisfies

$$i_J \leq r \cdot |J| = r^2, \tag{24}$$

since each group in $J$ hosts $r$ types.

Now, to verify if wipe-out statistics satisfy Barbour & Eagleson (1983, Thm. 1), let

$$k = k_N(x) := \lfloor x\, N^{1-1/r} \rfloor, \qquad x \geq 0 \text{ fixed}. \tag{25}$$

Then, as $k/N = xN^{-1/r}(1 + o(1))$ and, from Eq. (17),

$$p_k = \left(\frac{k}{N}\right)^r (1+o(1)) = \frac{x^r}{N}(1+o(1)), \quad \lambda_k = Np_k \to x^r. \tag{26}$$

We now check the six sufficient conditions of (Barbour & Eagleson, 1983, Thm. 1):

(i)  $Np_k \to x^r$. (26)

(ii)  $(N^{-1} \max i_J \leq r^2/N \to 0$. (24)

(iii)  $N^{1-r} \to 0$ for fixed $r \geq 2$.

(iv)  $C_1 q_1 \to 0$. (23) (21),

$$C_1 q_1(k) = Nr(r-1)\left(\frac{k}{N}\right)^{2r-1}(1 + o(1))$$
$$= r(r-1)\,x^{2r-1}\,N^{1-(2r-1)/r}(1 + o(1))$$
$$= \mathcal{O}\left(N^{1/r-1}\right) \to 0.$$

(v)  $N^2 q_1 N^{-1} \to 0$. (21)

$$Nq_1(k) = Nq_1(k) = N\left(\frac{k}{N}\right)^{2r-1}(1 + o(1))$$
$$= x^{2r-1}N^{1-(2r-1)/r}(1 + o(1)) = O\left(N^{1/r-1}\right) \to 0.$$

(vi)  $N|q_0 - p_k^2| \to 0$. (22) (17) (26)

$$\frac{q_0(k)}{p_k^2} = \frac{(k)_{2r}(N)_r^2}{(N)_{2r}(k)_r^2} = \frac{(k-r)_r}{(k)_r} \cdot \frac{(N)_r}{(N-r)_r}$$
$$= 1 + \mathcal{O}\left(\frac{1}{k} + \frac{1}{N}\right),$$

and for $k \to \infty$,

$$N|q_0 - p_k^2| = N \cdot \mathcal{O}(N^{-2}) \cdot \mathcal{O}\left(\frac{1}{k} + \frac{1}{N}\right) \to 0.$$

All conditions of (Barbour & Eagleson, 1983, Thm. 1) are satisfied, therefore $W_{k_N(x)} \Rightarrow \text{Poisson}(x^r)$, and

$$P_{k_N(x)} = \Pr\{W_{k_N(x)} = 0\} \to e^{-x^r}. \tag{27}$$

Using Eq.-(27) and a Riemann-sum argument with mesh $N^{-(1-1/r)}$,

$$\frac{\mu(N, r)}{N^{1-1/r}} = \sum_{k=0}^{N-1} \frac{1}{N^{1-1/r}} P_k \longrightarrow \int_0^\infty e^{-x^r}\, dx. \tag{28}$$

Finally, with the substitution $u = x^r$ (so $dx = \frac{1}{r}u^{1/r-1}du$),

$$\int_0^\infty e^{-x^r}\, dx = \frac{1}{r}\int_0^\infty e^{-u}u^{1/r-1}\, du = \frac{1}{r}\Gamma\left(\frac{1}{r}\right). \tag{29}$$

Therefore,

$$\mu(N, r) = \frac{\Gamma(1/r)}{r}N^{1-1/r}(1 + o(1)), \tag{30}$$

which proves Thm. (4.1). □

## B.3. Proof of Theorem 4.2

*Proof.* Let $q_k \in \{0, 1\}$ denote the indicator that patch compute is needed on the $(k+1)^{\text{th}}$ failure that takes the count from $k$ to $k+1$. We approximate the average overhead before wipe-out by averaging over the per-step stacks as:

$$S(N, r) \approx \frac{1}{\mu} \sum_{k=0}^{\lfloor \mu \rfloor - 1} \Big( \mathbb{E}[S(U_k)] + \mathbb{E}[q_k] \Big), \qquad (31)$$

where $\mu = \mathbb{E}[F]$ is the average failure count to the first wipe-out of Thm. (4.1).

First, we show $\mathbb{E}[S(U_k)] \approx c(k)$ with Hall's Marriage Theorem (Hall, 1987). What we want to show is that for failure count $k$ that $k \leq \mu(N, r)$, the strict inequality $S(U_k) > c(k)$ occurs with negligible probability.

Fix $s := c(k)$ and form the bipartite incidence graph between types and surviving groups: type $i$ is adjacent to group $w$ iff group $w$ is surviving and also hosting type $i$ ($w \in H_i \cap U_k$). Allowing *free* reordering within each group means that a group can be assigned up to $s$ types over the first $s$ stacks. Equivalently, replicate each group $w \in U_k$ into $s$ distinct copies $(w, 1), \ldots, (w, s)$ and set

$$L_i := (H_i \cap U_k) \times [s] \subseteq U_k \times [s],$$

the set of all feasible slots to which type $i$ may be assigned (one of its surviving hosts, at one of the first $s$ stacks). Then $S(U_k) \leq s$ is equivalent to the existence of a complete system of distinct representatives (C.D.R.) for $\{L_i\}_{i \in [N]}$; mathematically explained, an injection that can be written as $\phi : [N] \to U_k \times [s]$ such that $\phi(i) \in L_i$ for all $i$.

By Hall's Marriage Theorem (Hall, 1987), such an injection exists if and only if for every $E \subseteq [N]$,

$$|E| \leq \Big| \bigcup_{i \in E} L_i \Big| = s \cdot \Big| \bigcup_{i \in E} (H_i \cap U_k) \Big|. \qquad (32)$$

Therefore $S(U_k) > s$ implies that (32) fails for some witness $E$. Let the set of groups hosting the subset $E$ of shard types as $R$:

$$R := \bigcup_{i \in A} (H_i \cap U_k) \subseteq U_k, \qquad b := |R|.$$

Then $|E| > s|R|$ and every $i \in E$ satisfies $H_i \cap U_k \subseteq R$. Define the number of shards that *must* be computed by groups in $R$ as $C_R(k)$:

$$C_R(k) := \big| \{ i \in [N] : H_i \cap U_k \subseteq R \} \big|.$$

We have the implication

$$\{S(U_k) > s\} \subseteq \bigcup_{R \subseteq U_k} \{C_R(k) > s|R|\}, \qquad (33)$$

which means that when $R$ is *overloaded* with too many number of shards ($C_R(k)$) it is mandated to compute, then Eq. (32) is to be violated hence $S(U_k) > c(k)$.

Now we bound the RHS in Eq. (33) to formulate the cases where Eq. (32) is violated. Write $x := k/N$ and recall that $|H_i| = r$ and, by Lem. (B.2), $t_{\max} \leq 1$.

*Case 1. Single-group pinned overload ($b = 1$).* First case $S(U_k) > c(k)$ comes true (Eq. (32) violated), is when a group is mandated to compute certain shards; let us call it as the single-group pinned overload $O_w(k)$: For $w \in U_k$ define the pinned load

$$O_w(k) := \big| \{ i : H_i \cap U_k = \{w\} \} \big|.$$

If $O_w(k) \geq s + 1$ ($s := c(k)$), then $R = \{w\}$ violates Eq. (33) hence Eq. (32). Conditional on $w \in U_k$, for any type $i \in T_w$ (group $w$'s type set), $i$ is pinned to $w$ exactly when the other $r - 1$ hosts of $i$ all fail, hence

$$q_{k,N} := \Pr(H_i \cap U_k = \{w\} \mid w \in U_k)$$

$$= \frac{(k)_{r-1}}{(N-1)_{r-1}} = x^{r-1}(1 + o(1)) \quad (k < N). \quad (34)$$

From Lem. (B.2) ($t_{\max} \leq 1$), a binomial tail bound for $O_w(k)$ is valid and a union bound over at most $N$ groups gives

$$\Pr\big(\exists w \in U_k : O_w(k) \geq s + 1\big) \leq N \binom{r}{s+1} q_{k,N}^{s+1}. \qquad (35)$$

On the first-wipe-out scale $k \asymp \mu = \mathcal{O}(N^{1-1/r})$, we have $x \asymp N^{-1/r}$ and hence $q_{k,N} \asymp N^{-(r-1)/r}$, so the RHS of (35) behaves like $N^{1-(s+1)(r-1)/r}$ up to constants, which vanishes for all fixed $r \geq 2$ and $s = c(k) \geq 2$. Thus *Case 1* is negligible.

*Case 2. Multi-group Hall witnesses ($b \geq 2$).* Overload may not only happen to a single group, but on multiple groups as well. Fix $R \subseteq U_k$ with $|R| = b \geq 2$. A type counted in $C_R(k)$ must have all its surviving hosts inside $R$. Such a type either (i) has exactly one surviving host in $R$ (it is pinned to some $w \in R$), or (ii) has at least two surviving hosts in $R$. Hence,

$$\mathbb{E}[C_R(k)] \leq rb \cdot q_{k,N} + \binom{b}{2} \cdot \frac{(k)_{r-2}}{(N-2)_{r-2}}. \qquad (36)$$

The first term counts pinned types across the $rb$ types hosted by $R$. For the second term, note that if two groups shared two distinct types, it contradicts $t_{\max} \leq 1$; hence each group-pair in $R$ can jointly host at most one type, and for such a type to have all other $r - 2$ hosts failed has probability at most $(k)_{r-2}/(N-2)_{r-2}$.

For $k \leq \mu$ we have $x = \mathcal{O}(N^{-1/r})$, so both $q_{k,N}$ and $(k)_{r-2}/(N-2)_{r-2}$ vanish polynomially in $N$, and in particular $\mathbb{E}[C_R(k)] = o(b)$ uniformly over $b \leq m$. Since

$C_R(k)$ is a sum of indicators under sampling without replacement, a Chernoff bound (Chernoff, 1952; Hoeffding, 1963) applies and yields

$$\Pr\left(C_R(k) \geq sb\right) \leq \left(\frac{e\,\mathbb{E}[C_R(k)]}{sb}\right)^{sb} = N^{-\Omega(b)}. \tag{37}$$

Union bounding Eq. (37) over all $R \subseteq U_k$ and all $b$ implies

$$\Pr\left(S(U_k) > s\right) = o(1) \qquad \text{for all } k = 0, 1, \ldots, \lfloor \mu \rfloor - 1.$$

Together with the deterministic lower bound $S(U_k) \geq s = c(k)$, we obtain

$$\mathbb{E}[S(U_k)] = c(k) + o(1) \qquad (k \leq \mu). \tag{38}$$

Therefore, we show $\mathbb{E}[S(U_k)] \approx c(k)$. Now we compute the expectation value of the number of patch computes.

Fix $k$ and set $s := c(k)$, and set $n_k$ as

$$n_k = c(k)(N - k), \tag{39}$$

the number of stacks of computation before the all-reduce attempt. Let $d_i$ be the number of how many type $i$ appears among the $n_k$ slots; then $d_i \geq 1$ and $\sum_{i=0}^{N-1} d_i = n_k$. Let $u_k := |\{i : d_i = 1\}|$ be the number of singleton types: types that appear exactly once among the $n_k$ slots. Since every non-singleton types appear in $n_k$ slots at least 2,

$$n_k = \sum_i d_i \geq u_k + 2(N - u_k) = 2N - u_k$$
$$\implies u_k \geq \max\{0, 2N - n_k\}.$$

As each singleton type corresponds to exactly one slot, when the newly failed group is removed, the $s$ slots computed by that group in the current step are lost, and patch-compute is needed iff at least one of those lost slots was singleton.

Define the singleton-slot fraction as

$$\rho_k := \frac{\max\{0, 2N - n_k\}}{n_k}. \tag{40}$$

Under SPARe's minimal movement reordering by MCMF algorithm (Goldberg & Tarjan, 1990), the singleton slots are typically packed, correlated across groups rather than scattered independently across the $s$ slots of each group. Hence the probability that a uniformly random newly failed group hits at least one singleton is well-approximated to first order by the singleton-slot fraction itself:

$$\Pr(\text{patch compute at failure } k) \approx \rho_k. \tag{41}$$

Therefore, substituting (38) and (41) into (31) yields

$$\overline{S}(N, r) \approx \frac{1}{\lfloor \mu \rfloor} \sum_{k=0}^{\lfloor \mu \rfloor - 1} (c(k) + \rho_k),$$

with $n_k = c(k)(N - k)$ and $\rho_k = \max\{0, 2N - n_k\}/n_k$, proving Thm. (4.2). $\square$

## B.4. Proof of Theorem 4.3

*Proof.* Recall that the normalized time-to-train of SPARe+CKPT is Eq. (7)

$$J(r) := \frac{\text{time-to-train}}{T_0} = \frac{\overline{S}(N, r)}{A^\star(\mu(N, r)\, m)} \tag{42}$$

where $m$ is the MTBF between node failures, $\mu(N, r)$ is the average failure count to first wipe-out from Thm. (4.1), and $A^\star(\cdot)$ is the maximal availability obtained by the optimal CKPT period of Eq. (1),(2).

Note that the patch term in $\overline{S}(N, r)$ (5) is always bounded:

$$0 \leq 1 - (1 - \rho_k)^{c(k)} \leq 1, \tag{43}$$

from $0 \leq \rho_k \leq 1$. Therefore, the capacity term $\mathbb{E}[S(U_k)] \approx c(k)$ dominates $\overline{S}$, which increases as a step-function:

$$c(k) = 2 \text{ for } 1 \leq k \leq \frac{N}{2},$$
$$c(k) = 3 \text{ for } \frac{N}{2} < k \leq \frac{2N}{3},$$
$$c(k) = 4 \text{ for } \frac{2N}{3} < k \leq \frac{3N}{4},$$

and so on. Thus, as long as the wipe-out threshold $\mu(N, r)$ is below $N/2$, the $\mathbb{E}[S(U_k)]$ contribution in (5) stays pinned at $\approx 2$, and the only variation in $\overline{S}(N, r)$ comes from the bounded patch term (43). Once $\mu(N, r)$ exceeds $N/2$, a nontrivial fraction of indices $k$ in the sum (5) satisfy $k > N/2$, forcing $c(k) \geq 3$ and thereby increasing $\overline{S}(N, r)$ by an $\Omega(1)$ amount while the patch term is bounded $\leq 1$: the increase rate of the numerator of $J(r)$ (7): $\overline{S}(N, r)$ (5), jumps up after $k > \frac{N}{2}$.

Now we look at the denominator of $J(r)$ (7): $A^\star(\mu m)$ (2). Because $A^\star(T_f)$ increases monotonic in $T_f$, and $T_f = \mu(N, r)m$, increasing $r$ ($\mu(N, r)$ increases monotonic by $r$) improves the denominator in (7).

On the other hand, (5) shows that $\overline{S}(N, r)$ remains of constant order $\approx 2 + o(1)$ until the first capacity bump at $\mu(N, r) \approx N/2$, after which $\overline{S}(N, r)$ must increase because $c(k)$ becomes 3 for an increasing fraction of $k$. Therefore, the minimizer of $J(r)$ occurs at the *smallest* redundancy $r$ that pushes $\mu(N, r)$ to this first capacity transition,

$$\mu(N, r^\star) \approx \frac{N}{2}, \tag{44}$$

since below $r^\star$ one can still increase availability without paying the $c(k) \geq 3$ penalty on $\overline{S}(N, r)$.

Now we solve $\mu(N, r^\star) \approx \frac{N}{2}$ to find optimal redundancy $r^\star$ that minimizes $J(r)$. Let $\varepsilon := 1/r$ and rewrite $\mu(N, r)$ (3) as

$$\frac{\mu(N, r)}{N} = \Gamma(\varepsilon)\,\varepsilon\,N^{-\varepsilon}.$$

Imposing $\mu(N, r^\star) \approx \frac{N}{2}$ gives

$$\Gamma(\varepsilon)\,\varepsilon\,N^{-\varepsilon} \approx \frac{1}{2}. \tag{45}$$

For $\varepsilon \ll 1$, we use the standard expansion

$$\Gamma(\varepsilon) = \frac{1}{\varepsilon} - \gamma + \mathcal{O}(\varepsilon),$$

where $\gamma \approx 0.57721$ is the Euler-Mascheroni constant (NIST, 2025). Multiplying by $\varepsilon$ yields

$$\Gamma(\varepsilon)\,\varepsilon = 1 - \gamma\varepsilon + \mathcal{O}(\varepsilon^2).$$

Substituting into (45) and taking logarithms,

$$\log\!\big(1 - \gamma\varepsilon + \mathcal{O}(\varepsilon^2)\big) - \varepsilon\log N \approx -\log 2.$$

Using $\log(1 - x) = -x + \mathcal{O}(x^2)$ gives

$$-(\gamma + \log N)\,\varepsilon + O(\varepsilon^2) \approx -\log 2,$$

hence

$$\varepsilon \approx \frac{\log 2}{\log N + \gamma} \implies r^\star = \frac{1}{\varepsilon} \approx \frac{\log N + \gamma}{\log 2}$$
$$= \log_2 N + \frac{\gamma}{\ln 2}.$$

Finally, since $r$ is an integer, we take the nearest-integer (or floor) version:

$$r^\star \approx \left\lfloor \log_2 N + \frac{\gamma}{\ln 2} \right\rfloor \approx \big\lfloor \log_2 N + 0.833 \big\rfloor,$$

proving Thm. (4.3). □

## C. Monte-Carlo Simulation Results

*Table 4.* Simulation results for $N = 200$ over 1000 trials.

| $r$ | $\mu(N, r)$ | $\mu(N, r)$ | $\mathbb{E}[S(U_k)]$ | $\mathbb{E}[S(U_k)]$ |
|---|---|---|---|---|
| $r = 2$ | 12.5 | 13.2 | 1.84 | 1.90 |
| $r = 3$ | 30.5 | 31.3 | 1.93 | 1.96 |
| $r = 4$ | 48.2 | 49.8 | 1.97 | 1.98 |
| $r = 5$ | 63.6 | 65.3 | 1.96 | 1.98 |
| $r = 6$ | 76.7 | 78.5 | 1.97 | 1.99 |
| $r = 7$ | 87.8 | 89.7 | 1.97 | 2.00 |
| $r = 8$ | 97.1 | 99.3 | 1.99 | 2.03 |
| $r = 9$ | 105.1 | 106.9 | 2.03 | 2.07 |
| $r = 10$ | 112.0 | 113.6 | 2.09 | 2.11 |
| $r = 11$ | 118.0 | 120.9 | 2.14 | 2.16 |
| $r = 12$ | 123.2 | 126.3 | 2.17 | 2.20 |

*Table 5.* Simulation results for $N = 600$ over 1000 trials.

| $r$ | $\mu(N, r)$ | $\mu(N, r)$ | $\mathbb{E}[S(U_k)]$ | $\mathbb{E}[S(U_k)]$ |
|---|---|---|---|---|
| $r = 2$ | 21.7 | 22.5 | 1.89 | 1.94 |
| $r = 3$ | 63.5 | 65.3 | 1.97 | 1.98 |
| $r = 4$ | 109.9 | 108.9 | 1.97 | 1.99 |
| $r = 5$ | 153.3 | 154.6 | 1.99 | 1.99 |
| $r = 6$ | 191.7 | 194.8 | 1.99 | 2.00 |
| $r = 7$ | 225.1 | 227.2 | 2.00 | 2.00 |
| $r = 8$ | 254.0 | 254.9 | 1.99 | 2.00 |
| $r = 9$ | 279.1 | 281.4 | 2.00 | 2.02 |
| $r = 10$ | 301.1 | 302.3 | 2.00 | 2.04 |
| $r = 11$ | 320.4 | 324.8 | 2.05 | 2.08 |
| $r = 12$ | 337.4 | 340.0 | 2.10 | 2.11 |
| $r = 13$ | 352.5 | 355.3 | 2.14 | 2.15 |
| $r = 14$ | 366.1 | 366.8 | 2.17 | 2.18 |
| $r = 15$ | 378.2 | 382.1 | 2.20 | 2.21 |
| $r = 16$ | 389.2 | 393.4 | 2.22 | 2.25 |
| $r = 17$ | 399.2 | 400.6 | 2.24 | 2.27 |
| $r = 18$ | 408.3 | 412.6 | 2.28 | 2.31 |
| $r = 19$ | 416.6 | 420.2 | 2.31 | 2.33 |
| $r = 20$ | 424.2 | 426.4 | 2.34 | 2.36 |

*Table 6.* Simulation results for $N = 1000$ over 1000 trials.

| $r$ | $\mu(N, r)$ | $\mu(N, r)$ | $\mathbb{E}[S(U_k)]$ | $\mathbb{E}[S(U_k)]$ |
|---|---|---|---|---|
| $r = 2$ | 28.0 | 28.6 | 1.96 | 1.95 |
| $r = 3$ | 89.3 | 89.7 | 1.98 | 1.99 |
| $r = 4$ | 161.2 | 163.2 | 1.99 | 1.99 |
| $r = 5$ | 230.6 | 230.4 | 1.99 | 2.00 |
| $r = 6$ | 293.4 | 296.3 | 1.99 | 2.00 |
| $r = 7$ | 348.7 | 349.8 | 1.99 | 2.00 |
| $r = 8$ | 397.1 | 399.3 | 2.00 | 2.00 |
| $r = 9$ | 439.5 | 443.6 | 2.00 | 2.00 |
| $r = 10$ | 476.8 | 477.2 | 1.99 | 2.02 |
| $r = 11$ | 509.7 | 510.2 | 2.01 | 2.05 |
| $r = 12$ | 538.9 | 543.0 | 2.06 | 2.08 |
| $r = 13$ | 564.9 | 568.1 | 2.11 | 2.12 |
| $r = 14$ | 588.3 | 592.3 | 2.15 | 2.15 |
| $r = 15$ | 609.3 | 608.1 | 2.17 | 2.17 |
| $r = 16$ | 628.3 | 633.1 | 2.20 | 2.21 |
| $r = 17$ | 645.6 | 647.3 | 2.22 | 2.23 |
| $r = 18$ | 661.4 | 663.7 | 2.24 | 2.26 |
| $r = 19$ | 675.9 | 682.4 | 2.27 | 2.30 |
| $r = 20$ | 689.2 | 691.6 | 2.30 | 2.32 |
| $r = 21$ | 701.5 | 704.9 | 2.33 | 2.35 |
| $r = 22$ | 712.8 | 714.4 | 2.36 | 2.37 |
| $r = 23$ | 723.3 | 724.6 | 2.38 | 2.39 |
| $r = 24$ | 733.1 | 736.2 | 2.40 | 2.42 |
| $r = 25$ | 742.2 | 745.8 | 2.42 | 2.44 |
| $r = 26$ | 750.7 | 751.9 | 2.44 | 2.46 |

Table 4 ($N = 200$), Table 5 ($N = 600$), Table 6 ($N = 1000$) show the Monte-Carlo simulation results on the average failure count $\mu(N, r)$ (3), and the expectation value of all-reduce stack each scheme with $1,000$ trials. Columns of red color are the theoretical values from the formula Eq. (3) and Eq. (6), and the black colored are the simulation results. The results are highly consistent with Thm. (4.1) and Thm. (4.2): Across $N \in \{200, 600, 1000\}$ and all tested redundancies, Monte Carlo results match the closed-form formulas with Mean Absolute Percentage Error $1.13\%$ for $\mu(N, r)$, and $0.60\%$ for average all-reduce stack, with correlations $\geq 0.996$ (worst-case relative error $\leq 5.06\%$). See the file **reordering.ipynb** in Supplementary Materials for the code and the detailed simulation results. Simulation is implemented by emulating the trails of random independent failures and corresponding reordering events before the first wipe-out.

## D. Reordering Controller Algorithm

We detail how HK-FIXED and HK-FREE (Hopcroft & Karp, 1973), and MCMF (Goldberg & Tarjan, 1990) implement Alg. (2), reordering controller, and why their costs are negligible at $N \sim \mathcal{O}(10^{2-3})$.

**Bipartite feasibility model.** Define a bipartite graph with left vertices $\mathcal{L} = [N]$ (types) and right vertices $\mathcal{R} = U_k \times [S_A]$ (slots of computation among survivors). An edge $(i, (w, t))$ means survivor $w$ can compute type $i$ at stack $t$. A size-$N$ matching is equivalent to collecting all $N$ types by depth $S_A$ (one type per slot, every type covered).

**HK-FIXED (Phase 0: validate committed stacks).** Given the committed depth $S_A$ and the current per-group order $\text{stk}[w]$, construct edges $(\text{stk}[w][t], (w, t))$ for all $w \in U_k, t \leq S_A$. Run Hopcroft–Karp (Hopcroft & Karp, 1973) to test if the maximum matching has size $N$. If yes, all-reduce at $S_A$ succeeds without reordering; otherwise proceed to Phase 1.

**HK-FREE (Phase 1: find minimal $S(U_k)$ under free reordering).** For candidate $S = S_A, S_A + 1, \ldots, r$, allow free permutation within each group stack: add edges $(i, (w, t))$ iff $w \in H_i \cap U_k$. Run Hopcroft–Karp (Hopcroft & Karp, 1973); the first $S$ with matching size $N$ is $S(U_k)$. If no $S \leq r$ works, some type(s) has no surviving host (wipe-out) and the controller restarts.

**MCMF (Phase 2: minimum-movement reordering at $S(U_k)$).** At $S^\star := S(U_k)$, we seek a feasible assignment that changes stacks minimally. Use the same bipartite graph but assign each edge a movement cost as $0$ if already at position $t$, $1$ if making a movement. Compute a min-cost size-$N$ assignment via a standard min cost max flow al-

gorithm (Goldberg & Tarjan, 1990), then update each survivors' stack accordingly.

**Complexity and latency.** Hopcroft–Karp (Hopcroft & Karp, 1973) runs in $\mathcal{O}(E\sqrt{V})$ time where $E$ is the number of edges and $V$ is the number of vertexes (nodes). Here $V = N + (N - k)S$ and $E \leq N \cdot r \cdot S$, since each type has $\leq r$ surviving hosts and $S$ slots to be assigned per each group. For $N \sim \mathcal{O}(10^{2-3})$ and $r \leq \mathcal{O}(10^{0-1})$, and $S \approx 2\text{–}3$, these graphs are sparse and small, so HK-FIXED/HK-FREE are sub-100ms in compiled implementations. MCMF is run only upon failures and restricted to types and slots impacted by the failure, making its practical cost likewise negligible; we conservatively model the controller as 0.1s overhead per failure in our simulations.

**Potential Acceleration by Binary Search** The HK-FREE feasibility predicate is monotone in the candidate depth $S$: if all types are collectible by depth $S$, then they remain collectible for any larger depth $S' > S$. Therefore, the linear scan over $S$ in Alg. 2 can be replaced by a standard binary search over the ordered depth range, reducing the number of HK-FREE calls from $O(r)$ to $O(\log r)$ (Cormen et al., 2022). In our operating regime, however, the practical speedup is limited because the minimal feasible depth is typically small, around 2–3, and each HK-FREE call is already inexpensive at $N \sim 10^2\text{–}10^3$.

## E. Why Discrete-Event Simulation?

### E.1. Simulator Choice

Our simulator evaluates long-horizon fault-recovery dynamics of a large-scale training system, rather than only steady-state throughput under a fixed training configuration. The events to emulate include compute, gradient synchronization, failed all-reduce, failure detection, communicator shrink, reordering, checkpointing, and global restart. We therefore use a discrete-event simulator developed upon FedDES (Chen et al., 2025), a large parallel system simulation toolkit built on top of SimGrid (Casanova et al., 2013) which matches our system-level abstraction and fault-recovery evaluation target.

ASTRA-sim (Rashidi et al., 2020; Won et al., 2023) and Calculon (Isaev et al., 2023) are strong alternatives for ML-system performance modeling. ASTRA-sim is well suited for distributed-training SW/HW co-design and detailed performance exploration under specified model, parallelism, memory, and network configurations. Calculon provides a fast analytical model for LLM training performance and algorithm-system co-design. These tools mainly target inner-loop questions such as throughput, communication bottlenecks, and parallelism strategy. In contrast, SPARe requires an outer-loop reliability evaluation: how failures

accumulate and global restarts are bypassed, and how much wall-clock time is saved over a stochastic failure trajectory. FedDES/SimGrid is therefore a better fit for our evaluation target.

### E.2. SimGrid Validation and Use in HPC Studies

Prior work has validated SimGrid's modeling accuracy and scalability in distributed and HPC-relevant settings. Examples include flow-level TCP network models (Velho et al., 2013), scalable simulation of distributed applications and platforms (Casanova et al., 2014), MPI application simulation through SMPI (Degomme et al., 2017), and SimGrid-based data-transport simulation for in situ workflows (Suter, 2025). These works support SimGrid as an appropriate abstraction for system-level timing studies involving computation, communication, and resource contention.

SimGrid-based simulation has also been used as a main evaluation vehicle in peer-reviewed HPC venues. Examples include simulation-driven scheduling policies at SC (Carastan-Santos & De Camargo, 2017), volunteer-computing simulation at HPDC (Donassolo et al., 2010), bandwidth-aware scheduling at IPDPS (Beaumont & Rejeb, 2010), large-scale backfilling studies at CCGrid (Carastan-Santos et al., 2019), and edge devices system studies for federeated learning at SEC (Chen et al., 2025). This precedent supports our SimGrid-based discrete-event evaluation of SPARe, especially because direct experimentation at 100k–600k GPU scale is beyond realistic academic access.

## F. Additional Figures

### F.1. Communicator Topology

See Fig. 9.

### F.2. Simulation Flowchart

See Fig. 10.

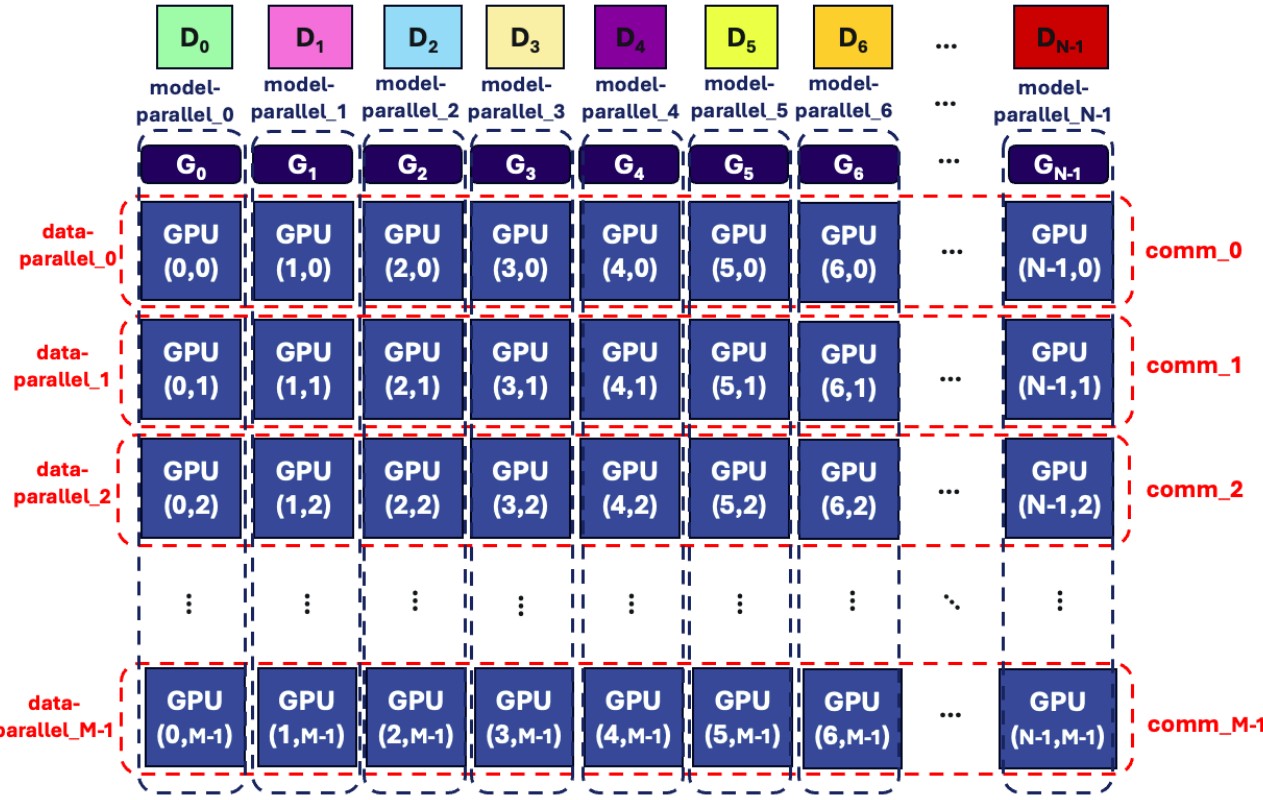

*Figure 9.* Communicator topology on data/model-parallel groups based on the descriptions of Sec. 2.1.

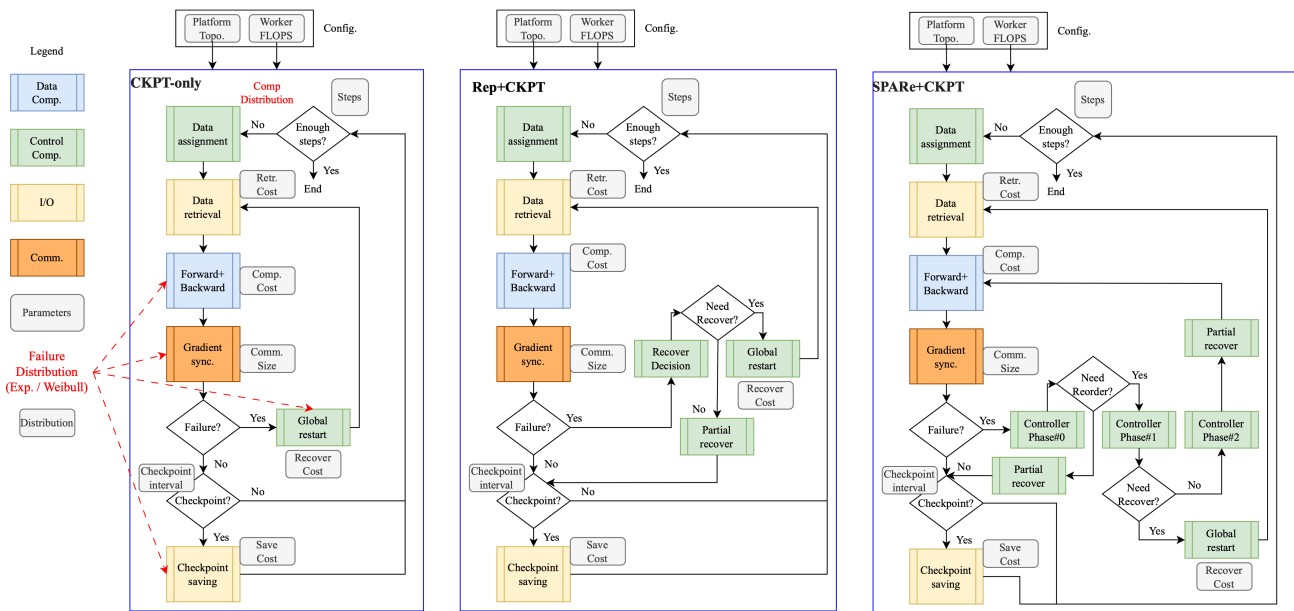

*Figure 10.* Simulation flowchart of CKPT-only, Rep+CKPT, and SPARe+CKPT

