# OpenReview forum: "SPARe: Stacked Parallelism with Adaptive Reordering for Fault-Tolerant LLM Pretraining Systems with 100k+ GPUs"
_ICML.cc/2026/Conference — ICML 2026 regular_

### Official Review · Reviewer_dF4J · 2026-03-09

**Soundness:** 2
**Presentation:** 2
**Significance:** 3
**Originality:** 3
**Overall Recommendation:** 4
**Confidence:** 2

**Summary:**

- The paper introduces SPARe (Stacked Parallelism with Adaptive Reordering), a fault-tolerance framework designed for LLM pretraining systems operating at extreme scales of 100k+ GPUs.
- At this scale, the system enters a restart-dominant regime where frequent node failures and linearly scaling global collective initializations cause system downtime to dominate the wall-clock training time.
- To mitigate this, SPARe utilizes synchronous data parallelism to stack redundant data shards across parallelism groups using a Cyclic Golomb Ruler distribution. It employs Hopcroft-Karp and Min-Cost Max-Flow algorithms to dynamically determine the minimal number of redundant computation stacks required from surviving groups to collect all partial gradients.
- Through SimGrid discrete-event simulations of a projected 600k H100 GPU cluster, the authors demonstrate that SPARe reduces the time-to-train by 40-50% compared to traditional replication with checkpointing, while maintaining a near-constant computation overhead of 2-3x.

**Compliance With Llm Reviewing Policy:**

Affirmed.

**Final Justification:**

The paper SPARe is a fault tolerance LLM pretraining framework, fully simulated on unprecedented scale of cluster.
The authors state that it is a pioneering work to preemptively address a very important problem in upcoming scale. This kind of pioneering work is a double-edge sword: on a darker side, this scale is not practical for now and might still be useless in the future, considering physical constraints of scaling the cluster (e.g. energy) or decision not to build such large cluster due to practical constraint (e.g. failure rate as indicated in the paper), but on a bright side, this could be a reference of distributed training for larger scale in the future.

Even though the authors tried to address most of my concerns, the following concern makes me doubt the feasibility of the design.
In the final authors' reply, the authors mentioned that FT-HSDP "could be an example" of inner topology. While I acknowledge that SPARe is inner-topology agnostic, this answer impresses that the experimental simulation does not fully simulate the training with a specific inner topology used. This weakens the credibility of evaluation results in the paper.

While I've raised my score as most initially raised concenrs are addressed, at the same time, I lowered my confidence to ask to deprioritize my rate in the decision, because after reading the reviews, rebuttals, and reading the paper again, it turns out I lost confidence in my understanding to the paper (i.e, most theoretical analysis and hierarchical communication part).

**Key Questions For Authors:**

- Could you justify why the framework rejects elastic redistibution (partitioning a lost data shard across the N-1 surviving groups)? Are these specific systems-level constraints that necessitate this rigid design choice that enforces a minimum 2x compute overhead over dynamic load balancing?
- In section 5.1, the simulation sets the partial recovery (communicator shrinking) cost to 0.1 seconds. Given your premise that collective operations scale linearly to 600k GPUs, how can a 100ms shrink time be justified? If shrinking actually takes minutes at this extreme scale, how does that impact the 40-50% time-to-train reduction?
- In Section 3.2, the Reordering Controller (RECTLR) relies heavily on the Hopcroft-Karp (HK) and Min-Cost Max-Flow (MCMF) algorithms to minimize overhead, yet their actual bipartite graph formulations are relegated to Appendix D. Could you provide a more detailed algorithmic walkthough or formal definition of the bipartite graph construction directly in the main text? Explaining this clearly in the main body would significantly improve the presentation and reproducibility of the core method.

**Limitations:**

- The paper should cite or compare against recent SOTA fault-tolerance frameworks (e.g. Bamboo or reCycle). While the introduction dismisses existing methods as merely reducing rework, these recent works explicitely aim to avoid global restarts via elastic reconfiguration; ignoring them limits the operational bounds of the study and weakens the paper's positioning.
- The authors state that collective communication reinitialization dominates runtime, however, SPARe's contirbution does not seem to mitigate this issue. SPARe's contribution of computational shuffling relies entirely on an existing partial recovery (communicator shrinking) mechanism (Section 2.3). The authors should include how SPARe's unique design mitigates communication reinitialization overhead.
- The authors must explicitely discuss their structural decision of discrete data stacking, as they currently provide no rationale for rejecting fractional data redistribution (elastic load-balancing).

**Strengths And Weaknesses:**

$+$ The theoretical foundation is rigorous. Section 4 delivers a highly structured and clear theoretical analysis of the system's performance limits. The presentation logically walks through three critical operational bounds, providing exploicit closed-form derivations.

$+$ The apper clearly defines the imminent "restart-dominant" regime, strongly motivating the necessity of the proposed work.

$+$ Achieving a 40-50% reduction in time-to-train at extreme scales represents massive financial and energy savings, significantly extending the viable limits of system scaling.

$-$ The framework structurally enforces a minimum 2x compute overhead upon the first failure by treating data shards as indivisible, atomic units. The paper does not consider or compare against elastic load-balancing techniques (e.g. dynamic batch redistribution) that could partition a lost shard across the N-1 surviving groups to achieve a near-1x overhead. The lack of justification for this rigid discrete stacking design over fractional compute redistribution limits the theoretical completeness of the architecture.

$-$ The evaluation is restricted to naive replication (Rep+CKPT) and standard checkpointing (CKPT-only) baselines. The authors justify this by making a generalized claim that existing fault-tolerance frameworks only minimize rework and still incur global restarts. Hoever, the paper fails to cite, discuss, or compare against highly relevant recent SOTA works (e.g. [Bamboo (NSDI'23)](https://www.usenix.org/conference/nsdi23/presentation/thorpe) or [reCycle (SOSP'24)](https://dl.acm.org/doi/abs/10.1145/3694715.3695960)) that explicitely utilize elastic pipeline and network reconfiguration to avoid global NCCL teardowns. Failing to acknowledge and analytically contrast SPARe against these modern elastic recovery methods severely limits the validation of its superiority in the targeted regime.

$-$ SPARe's practical applicability is limited to solely to currently non-existent extreme scales (10T model on 600k GPU cluster). In more realistic scale used for training (e.g. 8k-64k GPUs; even the Llama 405b model was trained on 16k GPU cluster), SPARe imposes a constant 2-3x computation overhead upon the first failure, it may perform significantly worse than standard checkpointing at scales where MTBF is longer and restart costs are lower.

$-$ The evaluation relies on a highly optimistic and contradictory assumption that communicator shrinking (partial recovery) takes only 0.1 seconds for a 600k GPU cluster. In section 2.3, the authors themselves cite prior work stating that shrinking takes "tens of milliseconds for hundreds of nodes". Given their core premise that collective operations scale linearly, assuming this operation takes only 100 milliseconds across 600k GPus directly contradicts their own cited basedline. If this cost is significantly higher at extreme scales, it could severely degrade the reported time-to-train gains.

---

> ### Author Rebuttal · Authors · 2026-03-30
>
> We sincerely thank the reviewer for the insightful and constructive feedback. Below we address all raised concerns one by one.
>
> ## Weaknesses
>
> **[W1: Elastic redistribution]** We fully agree that elastic redistribution is a notable design space. However, it is **costly and heavily constrained**, requiring substantial machinery to reroute lost work, rebalance execution, and synthesize a new schedule, often tightly coupled to model architecture and full-stack parallelism. At the extreme scale targeted by **SPARe**, inter-group data exchange would also impose massive communication. We therefore **intentionally choose discrete data stacking**, so inter-group communication is excluded and **SPARe** stays far more **model/parallelism-agnostic**.
>
> **[W2: Bamboo/ReCycle]** We fully agree that **Bamboo** and **ReCycle** are highly relevant and should be cited explicitly. We view them as **orthogonal and complementary** to SPARe rather than competing. **Bamboo** operates primarily at the **pipeline parallelism (PP)** layer, using redundant computation and pipeline bubbles to continue training under preemptions, whereas **SPARe** masks failure at data parallelism (DP) layer **above** that pipeline logic. **ReCycle** is a **hybrid DP/PP adaptation** system that reroutes micro-batches to DP peers, but it entails global restart when **any** DP group is lost; SPARe is designed to endure **multiple** group losses.
>
> **[W3: SPARe’s scale]** We fully agree that **SPARe** does not dominate at all scales. **SPARe** targets the restart waste, which was largely invisible before the 10k GPU scale **[1]** and has only recently emerged as a primary bottleneck over 100k+ GPUs **[2,3]**. We hope the reviewer will view this intentional focus not as a limitation, but as an attempt to study fault tolerance in an unprecedented and challenging regime.
>
> **[W4: Shrink cost]** We would like to clarify that the communicator **SPARe** shrinks is the **gradient synchronization communicator** across $N$ DP groups: an **$N$-rank communicator**, not one spanning all GPUs. Hence, in our simulations, shrink is for **N=200, 600, 1000**; exactly the scale **Bland et al. [4]** reported. We did state this in **Sec 2.3**, but we thank the reviewer for pointing out the potential confusion; we will clarify it again in **Sec 5.1**.
>
> ## Questions
>
> **[Q1: Elastic redistribution]** As discussed in **[W1]**, we intentionally exclude elastic redistribution as it is not lightweight: it requires substantial machinery and inter-group communication. Our goal is for **SPARe** to serve as a **universal failure-masking layer** for any training system that includes synchronous DP, so we deliberately avoid assumptions that would constrain that generality. We hope the reviewer will view this abstraction boundary as an intentional design choice for broad applicability.
>
> **[Q2: Shrink cost]** Please refer to **[W4]**. We would additionally note that, even under a pessimistic shrink cost of minutes scale, replacing a **60-min** global restart with a **few-min** shrink would still remain beneficial, although the absolute gain would of course be smaller.
>
> **[Q3: RECTLR description]** We fully agree. We will include a formal definition of bipartite graph construction and additional algorithmic details in **Sec 3.2**.
>
> ## Limitations
>
> **[L1: Bamboo/ReCycle]** We fully agree. We will explicitly cite **Bamboo** and **ReCycle**, and explain why we view them as **orthogonal and complementary** to **SPARe** rather than competing.
>
> **[L2: Restart cost scaling]** We would like to respectfully point out that **SPARe** **effectively** reduces restart waste **per failure** by masking them and thereby **bypassing many restarts**. For example, masking 200 failures is equivalent to reducing the restart waste to 0.5% (1/200) **per failure**. We therefore hope the reviewer will view this not as a limitation, but as a key strength of **SPARe**: it circumvents the subtle systems challenge of scaling restart cost while achieving desirable system availability (>90%).
>
> **[L3: Discrete data stacking]** We fully agree. We will clarify the intention behind our abstraction boundary: to maximize **SPARe’s versatility.**
>
> ## Revisions
>
> With sincere appreciation to the reviewer, we will make the following revisions:
>
> 1. Explain discrete data stacking and include comparison to **Bamboo** and **ReCycle** in a new **Related Work** section.
> 2. Clarify the shrink cost setting again in **Sec 5.1.**
> 3. Enhance the description of RECTLR in **Sec 3.2.**
> 4. Emphasize SPARe’s versatility in **Sec 1.**
>
> With these revisions, we believe the manuscript will become substantially clearer and stronger, and we are grateful to the reviewer for prompting these improvements.
>
> [1] https://www.usenix.org/conference/nsdi24/presentation/jiang-ziheng
>
> [2] https://arxiv.org/abs/2510.20171
>
> [3] https://arxiv.org/abs/2602.00277
>
> [4] https://ieeexplore.ieee.org/abstract/document/7152602

---

> > ### Author Rebuttal · Reviewer_dF4J · 2026-04-03
> >
> > Thank you for thorough response. The writing is super clear. And I also appreciate the authors' effort to position this paper as a pioneering study on a non-existent system.
> >
> > However, very sadly, some points are still groundless and I would like to request the authors to provide additional information.
> > - W1/Q1: The authors said they intentionally exclude elastic redistribution as it is really costly. But how much it is expensive? As the paper targets an unprecedented scale of cluster, I assume no existing paper would every measure how expensive elastic redistribution would be in this scale. It might not that be terrible because all rerouting lost work, rebalancing execution, and synthesizing a new schedule are for sure very expensive but at the same time the entire cluster can parallelize these works. Can the authors provide some simulated experimental results how bad it would be?
> > - W4/Q2: I understand that if there are N=200,600,100 gradient synchronization communicator, 0.1s may make sense with the reference. But this brings more confusion to me about how the cluster physical/logical topology exactly look like.
> > Let's say we have N=200 DP groups. On 600k GPU cluster each DP has 3k GPUs, and there is one gradient synchronization communicator for each DP. What kind of parallelism is used for these 3k GPUs within a DP other than replication? And does this synchronization communicator aggregate all gradients from GPUs within its DP before all-reducing between aggregator? It would be nice if there is a figure that illustrates overall architecture. GPUs are grouped to DPs, and communicators are located here and there, etc. Currently the targeting non-existent system is not even illustrated in the paper and fully imaginary.

---

> > > ### Author Response · Authors · 2026-04-05
> > >
> > > We thank the reviewer's insightful acknowledgement, and faithfully answer to all the remaining concerns.
> > >
> > > # Parallelism terminology amendment
> > >
> > > We acutely realize that our terminology **group** / **DP group** mismatch with the seminal **Megatron [1]** convention and hence created grave confusion to the reviewer. We sincerely apologize and respectfully amend our terminology. As explained in **Sec 2.1** of **SPARe** manuscript, **“group” /** **“DP group”** are meant as **“model-parallel group”** in **Megatron** convention.
> > >
> > > - **“Group” / “DP group”** in **SPARe** manuscript (**Sec 2.1**) : “.. the system consists of $N$ identical GPU groups.. Each group holds an identical copy of the entire model..”
> > > - **“Model-parallel group”** (Megatron): A group of GPUs holding a full model, **equivalent** to **“group” / “DP group”** in **SPARe** manuscript.
> > > - **“Data-parallel group”** (Megatron): A group of matching-rank GPUs across model-parallel groups. Partial gradients are synchronized across this group.
> > >
> > > We revise **SPARe**’s system setting in **Sec 2.1** accordingly as
> > >
> > > ### ”System consists of $N$ model-parallel groups, each holding an identical model and inner topology. At gradient synchronization step, each data-parallel group aggregates partial gradients with a communicator of world size (rank) $N$.”
> > >
> > > # [Q2] On physical/logical topology
> > >
> > > We respectfully clarify that **SPARe** is formulated on top of a **Megatron-style hierarchical hybrid parallelism**, with an **outer synchronous DP layer** and **abstracted inner topology (TP/PP, etc.)** beneath it. Therefore, in the revised terminology aligned with **Megatron**, we answer to the reviewer’s questions on a 600k cluster of $N=200$ model-parallel groups, each of 3000 GPUs.
> > >
> > > - **[What kind of inner topology?]:**  Each model-parallel group encompasses **an arbitrary yet identical inner topology, complex and globally coordinated within the group** so that any node failure would inevitably break the entire group. A good example would be **FT-HSDP [2]**.
> > > - **[How are the communicators formulated?]:** There are **3000 communicators**, each of **world size (rank) $N=200$**, aggregating partial gradients **across each data-parallel group** at the gradient synchronization. Therefore, upon failures, the communicators shrink from **at most 200 of ranks**, hence our **shrink cost setting of 0.1s** is reasonable and even conservative, assuming parallel execution.
> > > - **Figure 8 of Megatron [1]** and **Figure 3 of FT-HSDP [2]** are good illustrations of such model/data-parallel topology.
> > >
> > > Again, we sincerely apologize for the reviewer’s confusion, and **we fully agree that we need a good topology illustration of our own** in our manuscript. We will include an intuitive figure similar to **Figure 8 of [1]** and **Figure 3 of [2]** in the manuscript.
> > >
> > > [1] Megatron: https://arxiv.org/abs/1909.08053
> > >
> > > [2] FT-HSDP: https://arxiv.org/abs/2602.00277
> > >
> > > # [Q1] On the cost of elastic redistribution
> > >
> > > We fully agree that concrete numbers would strengthen the discussion. We would like to provide back-of-the-envelope projections on 600k GPUs for **ReCycle [3]** and **FT-HSDP [2]** based on their empirical scaling trend.
> > >
> > > **ReCycle [3]:** Their optimized **Planner** costs **278 / 942 / 3153 s** on **512 / 1024 / 2048 GPUs** when generating schedules for at most 25% failure accumulation, which shows a **power-law** scaling of **3.4x per doubling GPUs.** Projecting on **600k** GPUs, **just the planning alone** would take
> > >
> > > $ 3153s \times (3.4)^{8.2} \approx$ **2.28 years**.
> > >
> > > **FT-HSDP [2]:** They report reconfiguring from one failure costs **17 / 200 s** on **16k / 98k** GPUs, and along with another Meta optimization of **NCCLX [4]**, where the communication recovery costs **55.7 / 265 s** on **32k / 96k** GPUs, Meta optimization engineering shows a consistent **power-law scaling** of exponent 1.4 in the recovery cost. Projecting on the 600k GPUs, **reconfiguring** would cost
> > >
> > > $ 200s \times (\frac{600k}{98k})^{1.4} \approx$ **42 mins per failure**
> > >
> > > where failures are expected to occur **every 5 mins** at 600k. By contrast, **SPARe** needs only **sub-second communicator shrink and reordering per failure**, with the cost of 2~2.8x computation overhead.  We regretfully suggest that faithfully simulating either **Recycle** or **FT-HSDP** would require significant developing which would deserve a project of its own. However, at current stage of optimization, we respectfully suggest that elastic distribution indeed looks unpromising at 600k GPUs.
> > >
> > > # Revisions
> > >
> > > With sincere appreciation to the reviewer, we will make the following revisions:
> > >
> > > 1. Revise terminology throughout the paper to align with **Megatron** convention.
> > > 2. Add topology illustration in **Sec 2**
> > > 3. Add the **ReCycle/FT-HSDP** projection in **Appendices.**
> > >
> > > We are deeply grateful to the reviewer for prompting these improvements.
> > >
> > >
> > >
> > > [3] ReCycle: https://arxiv.org/abs/2405.14009
> > >
> > > [4] NCCLX: https://arxiv.org/abs/2510.20171

---

### Official Review · Reviewer_CwXF · 2026-03-11

**Soundness:** 2
**Presentation:** 3
**Significance:** 2
**Originality:** 3
**Overall Recommendation:** 2
**Confidence:** 4

**Summary:**

The paper proposes SPARe, a fault-tolerance framework that adapts to frequent failures in large-scale LLM pretraining. SPARe minimizes the overhead associated with failure recovery by stacking redundant data shards across parallelism groups and adaptively reordering execution. This approach is shown to reduce time-to-train by 40-50% in simulation compared to traditional replication strategies while maintaining availability.

**Compliance With Llm Reviewing Policy:**

Affirmed.

**Final Justification:**

I would maintain my score as the discrepancy between simulator and real-world training is great enough that I feel the paper cannot be accepted in its current form.

**Key Questions For Authors:**

Q1. What potential impact could multi-tiered storage have on SPARe's performance in real-world deployments? For instance, CPUs can be leveraged to provide redundancy for fault tolerance and rapid recovery (e.g., GEMINI [SOSP’23]). The loss of a GPU shard does not necessarily equate to complete data loss (i.e., the “wipe-out” in the paper).

Q2. Can the SPARe framework be adapted to systems with a higher degree of heterogeneity, such as varying failure rates across different GPUs? How would the algorithm address these complexities?

Q3. Real-world system failures are often more complex than simple GPU failures (e.g., Llama3 technical report, ByteRobust [SOSP’25]). Node-level failures or communication link failures (e.g., switches/buses) could result in multiple GPU failures simultaneously. How does SPARe handle such scenarios and modeling such failure patterns?

Q4. Are there any real-world traces or deployment experiences with SPARe that could be shared to provide further insight into its practical applications?

**Limitations:**

The paper would benefit from a deeper exploration of real-world challenges and a discussion on the limitations of deploying SPARe at the scale of 600k GPUs in practical environments.

**Strengths And Weaknesses:**

S1. The research addresses an urgent issue in large-scale machine learning systems.

S2. The use of adaptive reordering in SPARe is a novel way to reduce computation overhead while masking failures.

S3. The paper offers strong theoretical results, including closed-form expressions for failure count and computation overhead.

---

W1. The paper overlooks multi-tiered storage, assuming fault tolerance is limited to GPUs. This narrow scope misses the potential benefits of incorporating recovery mechanisms from multi-tiered storage, which is more reflective of real-world scenarios.

W2. The assumption of a Poisson distribution for failure statistics may not be applicable in real-world systems, where failure rates can be more complex. The inclusion of real-world traces from large GPU clusters would strengthen the findings. Several prior works, such as Bamboo [NSDI’23], have suggested that failures may follow specific patterns (e.g., GPUs in the same zone), making it important to validate the proposed models with actual data.

W3. While SPARe+CKPT demonstrates promising results in simulations, deploying it at the scale of 600k GPUs may present unforeseen challenges. The paper could be further refined by implementing a real system and assessing its practical benefits.

---

> ### Author Rebuttal · Authors · 2026-03-30
>
> We sincerely thank the reviewer for the insightful and constructive feedback. Below we address all raised concerns one by one.
>
> ## Weaknesses
>
> **[W1: Multi-tiered storage]** We view **multi-tiered storage** as **orthogonal and complementary** to **SPARe** rather than competing, since they address different layers of fault tolerance. Methods such as **GEMINI** primarily optimize the **checkpoint/rollback path**. By contrast, **SPARe** reduces how often the job must enter that recovery path in the first place, by masking failures at the data parallelism (DP) layer. Therefore, **SPARe** targets an overhead **prior** to the rollback process.
>
> **[W2: Realistic failures]** We respectfully clarify that the **Poisson approximation in Theorem 4.1 is not an assumption on physical node failure arrivals**. It is for the **wipe-out incident statistics** induced by **SPARe**’s least-overlapping shard placement. Also, our simulations already use the realistic **Weibull** failure distribution. Moreover, **SPARe** is formulated at **DP group granularity:** correlations **within** a group are abstracted into a single group failure. This abstraction is also consistent with **ByteRobust,** which places protection outside the same parallel-group fault domain, and **Bamboo,** which places mutually dependent nodes across different availability zones to reduce correlated failures.
>
> **[W3: Real system]** We fully agree that a real system implementation of **SPARe** would be valuable. At the same time, we would like to respectfully note that **SPARe** targets the extreme scale of **100k+ GPUs**, which is not realistically accessible in a standard academic environment. The present paper is therefore intended as a **theoretical foundation** for a failure masking scheme at the DP layer for the imminent restart-dominant regime. Given the scope of our work, we believe **SimGrid-based discrete-event simulation** is appropriate rather than ad hoc: **SimGrid** has been repeatedly validated to track real system behavior at the abstraction level relevant to large-scale distributed/HPC studies **[1-4]**, and simulation-centered works using **SimGrid**, even without real system implementation, have been peer-reviewed and accepted in major venues such as **SC [5], HPDC [6], IPDPS [7], and CCGrid [8]**.
>
> ## Questions
>
> **[Q1: Multi-tiered storage]** As discussed in **[W1], multi-tiered storage** does not mitigate the global restart: it reduces **rollback cost**, while **SPARe** reduces **restart frequency**. We also clarify that **wipe-out** in **SPARe** does **not** mean irreversible data loss in storage; it means the active job can no longer collect all required shard types from surviving DP groups, and therefore must enter the restart + rollback path.
>
> **[Q2: Heterogeneous failures]** We would like to respectfully note that **SPARe**’s algorithmic behavior depends on **failure count** and **survivor set**, not on the temporal law of failures. Thus **Algorithm 1 and 2**, as well as the count-based results in **Theorems 4.1 and 4.2**, are unchanged under heterogeneity. What affected is the clock-time checkpointing optimization in **Theorem 4.3**, and optimal checkpointing period should be recalculated.
>
> **[Q3: Real failures]** As discussed in **[W2]**, we would like to respectfully note that the intra-group correlations are outside **SPARe**’s abstraction boundary, as the core logic is entirely formulated at DP layer. That said, even when a fault induces multiple group failures simultaneously, **SPARe** still operates on the resulting failed group set in the same way.
>
> **[Q4: Real system]** As discussed in **[W3], w**e do not yet have real world deployment traces for SPARe. We hope the reviewer will view this work as a theoretical foundation with real system implementation being the natural next stage of the research agenda.
>
> ## Revisions
>
> With sincere appreciation to the reviewer, we will make the following revisions:
>
> 1. Compare **SPARe** and **multi-tiered storage**, and cite **ByteRobust/Bamboo** to explain **SPARe**’s abstraction boundary in a new **Related Works** section.
> 2. Clarify in **Sec 3, 4** that **SPARe**’s DP-layer logics stay unchanged under failure heterogeneity, except for **Theorem 4.3**.
> 3. Discuss the validity and recognition of SimGrid by HPC community in **Sec 5.1**.
>
> With these revisions, we believe the manuscript will become substantially clearer and stronger, and we are grateful to the reviewer for prompting these improvements.
>
> [1] https://dl.acm.org/doi/abs/10.1145/2517448
>
> [2] https://www.sciencedirect.com/science/article/pii/S0743731514001105
>
> [3] https://ieeexplore.ieee.org/document/7855780
>
> [4] https://ieeexplore.ieee.org/stamp/stamp.jsp?tp=&arnumber=11186460
>
> [5] https://dl.acm.org/doi/abs/10.1145/3126908.3126955
>
> [6] https://dl.acm.org/doi/abs/10.1145/1851476.1851565
>
> [7] https://ieeexplore.ieee.org/stamp/stamp.jsp?tp=&arnumber=5470450
>
> [8] https://ieeexplore.ieee.org/abstract/document/8752660

---

> > ### Author Rebuttal · Reviewer_CwXF · 2026-04-03
> >
> > Thanks for the rebuttal. My concerns still remain.
> > - The failure pattern simulation (either Poisson approximation or Weibull failure distribution) can hardly be realistic.
> > - Regarding "the core logic is entirely formulated at DP layer", would there be notable resource waste if we must employ high model parallel degrees (particularly for large-scale models).
> > - Most of the cited papers are before 2020 (and half of them before 2015), and none of them focus on distributed training of LLMs over large-scale clusters. There are unforeseen challenges (e.g., storage, network, etc.) when tremendous GPUs are used, so it is unclear how the simulation represents real-world scenarios well.

---

> > > ### Author Response · Authors · 2026-04-05
> > >
> > > We thank the reviewer for the acknowledgement, we faithfully answer to all remaining concerns.
> > >
> > > # [Q1] On Poisson approximation
> > >
> > > We respectfully clarify that the Poisson approximation in **SPARe** is **NOT an individual node-level failure model**. It is about the **statistics of wipe-out**, an incident where a certain type of data shards become unavailable among surviving groups. It is a **statistical state** originating from **SPARe**’s shard distribution rule (**Appendix B.1**), which has any two shard types overlap in at most one group; in this way, wipe-out incidents of different shard types are weakly dependent to each other so that they can be approximated to be statistically independent to each other.
> > >
> > > # [Q1] On Weibull distribution
> > >
> > > We respectfully clarify that **Weibull** is still a **legitimate standard proxy** to represent real-life failures **at the group-level abstraction** as shown in **Titan trace study (2025) [1]** and **REFT (2024) [2]**. At the same time, **we fully agree that it does not emulate the exact raw-node failure physics** at contemporary cluster scale, as the reviewer has insightfully pointed out. Our claim is that **Weibull** is a defensible and reasonable failure model when emulating real-life system behavior **at the abstraction level of parallel group granularity** where subtle correlations between individual node failures are abstracted into a single group failure. We respectfully list the recent evidences behind our claim as below:
> > >
> > > - **Titan trace study** **[1], Table 3 at page 21**: In their **7 years trace** over **30,000 GPUs** in Cray XK7 Titan, they show that **Weibull distribution best fits** in terms of leave-one-out cross-validation information criterion (**LOOIC**) among their failure model pool.
> > > - **REFT [3], Sec 5 Fig 8:** They study a Megatron-style distributed ML system of 3072-devices, and evaluated their system reliability on **Weibull distribution.**
> > >
> > > **We fully agree that no failure model can faithfully predict real-life failure traces in an unprecedented 600k GPU cluster.** At the same time, we respectfully suggest that **under the parallel-group level abstraction**, which is empirically supported by **Bamboo** and **ByteRobust** where they showed individual node correlations can be invisible across the parallel-groups with engineering, **Weibull distribution** may still be a legitimate proxy for system evaluation.
> > >
> > > # [Q2] On high model parallel degrees
> > >
> > > Since data parallelism is the outmost layer in contemporary Megatron-style hierarchical hybrid parallelism, and since **SPARe** is orthogonal to the inner-group topology such as Tensor Parallelism (TP) and Pipeline Parallelism (PP), we respectfully clarify that the high model parallel degrees would not incur notable resource waste in principle. What would exist **under a fixed GPU budget** however, is a **trade-off between the parallel degree and the number of groups**; higher parallelism degree would mean fewer groups since each group would require more GPUs. As shown in **Sec 5**, **SPARe**’s performance is **expected to be better at larger number of groups** since that would lower the probability of wipe-out.
> > >
> > > # [Q3] On unexpected challenges in real-life
> > >
> > > We fully agree with the reviewer that the practical impact of **SPARe** can only be established through a real-world implementation, and that deploying **SPARe** in an actual large-scale training system would inevitably surface many unexpected systems challenges. At the same time, our goal in this work is purely academic in nature: to establish the **theoretical foundation of a new fault-tolerance framework** for the imminent restart-dominant regime. We hope the reviewer would not view **SPARe** as a deployment paper, but as a **principled proposal** that identifies a previously unexplored design space, derives its core properties formally, and validates them in a controlled simulations.
> > >
> > > We respectfully believe this is a meaningful contribution for academia. Many important systems ideas are **first introduced as theoretical or simulation-based studies** precisely because the target regime is not yet practically accessible to most researchers, or because building the full production infrastructure is beyond the scope of a first paper. In such cases,  we believe academia’s role is to ask whether a new mechanism is possible in principle, to characterize its properties rigorously, and to provide a foundation that future implementation work can build on. We position **SPARe** in this spirit: as an analytically grounded step toward future fault-tolerant training systems, rather than as an immediate production-ready protocol.
> > >
> > > We are grateful to have this opportunity to answer to the reviewer's concerns. We hope we have adequately resolved them.
> > >
> > > [1] Titan trace study: https://www.tandfonline.com/doi/abs/10.1080/00401706.2025.2475783
> > >
> > > [2] REFT: https://arxiv.org/abs/2310.12670
> > >
> > > [3] FT-HSDP: https://arxiv.org/abs/2602.00277

---

### Official Review · Reviewer_X7Jp · 2026-03-12

**Soundness:** 3
**Presentation:** 3
**Significance:** 3
**Originality:** 3
**Overall Recommendation:** 4
**Confidence:** 2

**Summary:**

The paper targets the “restart-dominant” regime in 100k+ GPU LLM pretraining, where failures are frequent and global restarts/collective re-initialization can dominate wall-clock time.  It proposes SPARe, which stacks redundant data shards across data-parallel groups and, upon failures, adaptively reorders which shard-stacks get computed so that all partial gradients remain collectible with near-constant ~2–3\times compute overhead (instead of the r\times overhead of degree-r replication).

**Compliance With Llm Reviewing Policy:**

Affirmed.

**Key Questions For Authors:**

See strengths and weaknesses.

**Strengths And Weaknesses:**

The paper is well motivated in the sense that in current large scale training, restarts (init/sync) dominate at  100k–600k GPU scale. Therefore, reducing global starts matter. I liked how the authors provide an explicit training loop + reordering controller pipeline (HK-FIXED, HK-FREE, MCMF) and define failure/wipe-out concepts clearly. One draw back might be that they conduct simulations instead of an actual training setup. However, the evaluation is set up like an HPC paper: system parameters table, discrete-event simulation, and comparison against replication. That said, I have several questions.

1. Is a “type” a data shard, a microbatch, a data-parallel rank’s shard, or something else?
2. HK-FREE can reorder which shard a group computes in stack positions 1..S. If optimizer state depends on data ordering (even subtly), are the authors assuming data order invariance?
3. Algorithm 2 searches incrementally for the minimal feasible depth. Could this be sped up via monotonicity + binary search?
4. Authors explicitly assume failures are detected only when the system calls all-reduce, and the failed all-reduce costs 0.5×Ta in expectation. How robust the results are if failure detection is earlier (ULFM-style) or later (timeouts), and how that changes patch compute and schedule feasibility?

---

> ### Author Rebuttal · Authors · 2026-03-30
>
> We sincerely thank the reviewer for the insightful and constructive feedback. Below we address all raised concerns one by one.
>
> ## Weakness
>
> **[W1: Real system]** We fully agree that a real system implementation of **SPARe** would be a natural and impactful next stage agenda. At the same time, we would like to respectfully note that **SPARe** targets the extreme scale of **100k+ GPUs**, which is not realistically accessible in a standard academic environment. The present paper is therefore intended as a **theoretical foundation** for a failure masking scheme at the **data parallelism (DP) layer**. Given the scope of our work, we believe **SimGrid-based discrete-event simulation** is appropriate rather than ad hoc: **SimGrid** has been repeatedly validated to track real system behavior at the abstraction level relevant to large-scale distributed/HPC studies **[1-4]**, and simulation-centered works using SimGrid, even without real system implementation, have been peer-reviewed and accepted in major venues such as **SC [5], HPDC [6], IPDPS [7], and CCGrid [8]**.
>
> ## Questions
>
> **[Q1: “Type”]** A **type** is the **indexed identity of a data shard** obtained from partitioning the original training workload. Thus, under $N$-way synchronous data parallelism, there are $N$ shard types, and under replication degree $r$, each type has $r$ replicas. We thank the reviewer for pointing out that this terminology should be defined more explicitly.
>
> **[Q2: Data order invariance]** We would like to respectfully clarify that **SPARe** does **not** assume data order invariance. **SPARe** does not change **which shard is computed** at a given optimizer step, nor the **optimizer state used** for that computation. It changes only **which surviving group supplies each required shard type** after failures, and thus only the **order in which partial gradients are gathered** at the synchronization step. Since the optimizer update is applied only after all required partial gradients have been collected into the same full gradient, the final update remains identical to the vanilla synchronous DP baseline.
>
> **[Q3: HK-FREE acceleration]** We thank the reviewer for this insightful suggestion. Yes, accelerating **Algorithm 2** via **monotonicity + binary search** is indeed viable in principle. The feasibility condition checked by **HK-FREE** is indeed monotone: if all $N$ types are collectible by depth $S$, they remain collectible for any larger depth. That said, in our operating regime the practical gain is likely small, since the feasible depth is typically only $2\sim3$, and **HK-FREE** iterations already take sub-second as discussed in **Appendix D**. Nevertheless, we recognize **monotonicity + binary search** as a useful implementation refinement, and deeply thank the reviewer for this insightful suggestion.
>
> **[Q4: Earlier/later failure detection]** We thank the reviewer for this important question. Earlier failure detection can only improve **SPARe**, because it reduces or even eliminates the need for **patch compute** after reordering; correspondingly, **Theorem 4.2** states that with sufficiently early detection, the computation overhead approaches the theoretical lower bound as the probability of patch compute converges to zero. On the other hand, if failures are surfaced only after longer timeout-based detection, then the failed all-reduce may take substantially longer than a successful one. This would certainly worsen the **absolute time-to-train** for **SPARe** as well as for any baseline, but it does **not** change SPARe’s core logic: **Algorithm 1** and **Algorithm 2** remain the same, and only the latency of the failed synchronization event becomes larger.
>
> ## Revisions
>
> With sincere appreciation to the reviewer, we will make the following revisions:
>
> 1. Clarify **type** in **Sec 2**.
> 2. Clarify that **SPARe** does **not** alter optimizer state or the resulting update in **Sec 3.2**.
> 3. Add to **Sec 3.2** the monotonicity of the minimal feasible stack search and the viability of **binary-search acceleration**.
> 4. Revise **Sec 4.1** around **Theorem 4.2** to clarify the effect of earlier / later failure detection.
> 5. Enhance **Sec 5** with additional discussion on the validity and recognition of SimGrid-based discrete-event simulation by HPC community.
>
> With these revisions, we believe the manuscript will become substantially clearer and stronger, and we are grateful to the reviewer for prompting these improvements.
>
> [1] https://dl.acm.org/doi/abs/10.1145/2517448
>
> [2] https://www.sciencedirect.com/science/article/pii/S0743731514001105
>
> [3] https://ieeexplore.ieee.org/document/7855780
>
> [4] https://ieeexplore.ieee.org/stamp/stamp.jsp?tp=&arnumber=11186460
>
> [5] https://dl.acm.org/doi/abs/10.1145/3126908.3126955
>
> [6] https://dl.acm.org/doi/abs/10.1145/1851476.1851565
>
> [7] https://ieeexplore.ieee.org/stamp/stamp.jsp?tp=&arnumber=5470450
>
> [8] https://ieeexplore.ieee.org/abstract/document/8752660

---

> > ### Author Rebuttal · Reviewer_X7Jp · 2026-04-04
> >
> > I thank the authors for clearly addressing my concerns. As such, I would like to keep my original score, leaning towards acceptance.

---

> > > ### Author Response · Authors · 2026-04-05
> > >
> > > We are very happy that the reviewer's concerns are all resolved, and also sincerely grateful for the reviewer's affirmation.

---

### Official Review · Reviewer_rb5Y · 2026-03-13

**Soundness:** 3
**Presentation:** 3
**Significance:** 2
**Originality:** 3
**Overall Recommendation:** 4
**Confidence:** 4

**Summary:**

Thank you for submitting to ICML. The paper targets failures in large scale LLM training that spans 100k GPUs. The paper states that restarting jobs from a checkpoint has high restart overheads and at large clusters, this can lead to the job spending more time on restarts than actual job progress due to frequent failures. Therefore, it is better to maintain replicas and replace the failed accelerators seamlessly with a replica. As the failure rates are high, it is impractical to maintain a corresponding degree of replication. So, SPARe proposes a mechanism that retians near constant replication overhead while providing high availability achieved by traditional replication. The main insight is to repartition the shards in a way to avoid redundant computations unless necessary.

**Compliance With Llm Reviewing Policy:**

Affirmed.

**Final Justification:**

The authors answered my questions in the rebuttal and the subsequent replies which clarified key details. My question is, how much does this NCCL group restart overhead contribute to failure recovery time. Any simulation or experiment answering this quesiton will be a very valuable addtion to the paper. I had to read the FT-HSDP paper to find out that it addresses the question I have in section 4.2 and section 6.1 using their own implementation of NCCL for synchronization All-reduce called FTAR (Fault tolerant All-Reduce). The paper mentions in 6.1 that it takes a few seconds for this implementation to change the group. I recommend the authors to mention this early in the paper to clearly articulate where SPARe stands in the stack and how other parts of the stack work with SPARe. I think most of the components are already there in FT-HSDP and SPARe is reducing the number of required replicas with guarantees.

**Key Questions For Authors:**

1. How does the paper compare to TrianMover? https://arxiv.org/abs/2412.12636
2. It would be nice to understand what is the parallelism of trianing jobs on GPUs within a DP group duing the simulation? Assuming a H100 DGX, it will be good to show GPU memory utilization during steady state and failure recovery.
3. How does the proposed fault tolerance scheme compare to maintaining a pool of idle GPUs? Seems like SPARe has the benefit that it doesn't necessitate any new network communication to transfer weights during the tolerable failures.
4. What is the difference between the current simulations and ML training simulators like Astra-sim or Calculon?

**Limitations:**

Yes

**Strengths And Weaknesses:**

[S]: A timely and relevant problem

[S]: Novel solution that measures fault tolerance by the expected random failures before first wipe-out for a given replication degree r. This scheme can tolerate a large number of failures with very few replicas.

[S]: The expected value is validated using Monte-carlo simulations

[W]: Evaluation is in a simulator that does not fully model the distributed training stack of ML.

[W]: The evaluation setup does not explain the parallelism plan within each DP group.

[W]: It will be nice to have a baseline modeling restarts.

---

> ### Author Rebuttal · Authors · 2026-03-30
>
> We sincerely thank the reviewer for the insightful and constructive feedback. Below we address all raised concerns one by one.
>
> ## Weaknesses
>
> **[W1: Full stack training]** We respectfully clarify that our evaluation is **intentionally not** on full stack training. **SPARe** is formulated at the **data parallelism (DP) layer**, and our goal is to evaluate the **long-horizon stochastic fault-recovery process** of training by modeling compute, collectives, failures, checkpointing, shrink, reordering, and restart as **block events** rather than micro-level execution inside a specific model stack. This abstraction is deliberate: **SPARe** is meant to provide a **model/parallelism-agnostic universal failure masking layer** above those lower-level system choices, assuming only synchronous DP.
>
> **[W2: Intra-group topology]** We fully agree this abstraction boundary should be clarified more clearly. In **SPARe**, each DP group is a fault-tolerance unit; whatever tensor/pipeline parallelism or other inner topology is abstracted into group-level system parameters. This is **intentional**: many recent fault-tolerance systems are tightly optimized for particular model structures or parallelism layouts **[1-6]**, whereas **SPARe** is designed to be a **universal failure masking layer** for any system with synchronous DP. We hope the reviewer will view this **versatility of SPARe** as a strength, not a limitation.
>
> **[W3: Restart baseline]** We respectfully clarify that **CKPT-only** is precisely the **restart baseline**: every detected failure triggers a restart protected only by checkpointing. **SPARe+CKPT** and **Replication+CKPT** use the same restart model; they differ only in that, before the inevitable restart after a system failure (wipe-out), restarts are replaced via shrink + reordering.
>
> ## Questions
>
> **[Q1: TrainMover]** We view **TrainMover** as **orthogonal and complementary** to **SPARe** rather than directly competing. **TrainMover** reduces the **cost** of each interruption via idle preheated GPUs (hot spares),  whereas **SPARe** reduces the **frequency** of interruptions by masking failures at the DP layer. In availability terms, **SPARe** increases system failure interval, while **TrainMover** reduces restart cost.
>
> **[Q2: Intra-group topology]** We respectfully believe GPU memory utilization inside a DP group is outside **SPARe**’s abstraction boundary. **SPARe** does not change intra-group model-state placement or GPU memory behavior when masking failures at the DP layer; it only changes **which groups continue contributing shard types** to the next synchronization. Our evaluation is therefore intentionally performed at **DP group granularity**.
>
> **[Q3: Hot spares]** **Hot spare methods** address a different layer of the fault-tolerance stack. **SPARe** reduces the interruption **frequency** by masking failures, whereas **hot spare methods** reduce the recovery **cost** once an interruption occurs. As the reviewer insightfully notes, replacing failed GPUs with hot spares still incurs nontrivial overhead, including data transfer and optimizer state migration. While cheaper than a global restart, this overhead can still be substantial at the **100k+ GPU scale** targeted by **SPARe**. Thus, even with hot spares, **SPARe** remains valuable by masking failures during the replacement; conversely, **hot spare methods** can replenish DP groups and delay wipe-out, which further benefits **SPARe**. We therefore view the two as **orthogonal and complementary** rather than competing.
>
> **[Q4: Simulator choice]** We respectfully distinguish these tools by evaluation target. **SimGrid** is designed for **application-agnostic** system behavior over a **long-horizon stochastic fault-recovery evaluation**, which matches **SPARe**. By contrast, **ASTRA-sim** and **Calculon** are better suited to evaluate a **specified training workload / steady-state execution configuration** and detailed inner-loop performance under a **largely repetitive training loop**, typically over **shorter horizons**.
>
> ## Revisions
>
> With sincere appreciation to the reviewer, we will make the following revisions:
>
> 1. Highlight the **model/parallelism-agnostic versatility** of **SPARe** in **Sec 1,3.**
> 2. Add **Related Work** section and discuss **hot spare methods**.
> 3. Revise **Sec 5** to better explain our use of **SimGrid** and clarify its distinction from **ASTRA-sim** and **Calculon**.
>
> With these revisions, we believe the manuscript will become substantially clearer and stronger, and we are grateful to the reviewer for prompting these improvements.
>
> [1] https://dl.acm.org/doi/abs/10.1145/3694715.3695960
>
> [2] https://arxiv.org/abs/2602.00277
>
> [3] https://dl.acm.org/doi/abs/10.1145/3731569.3764838
>
> [4] https://arxiv.org/abs/2407.04656
>
> [5] https://arxiv.org/abs/2504.06095
>
> [6] https://arxiv.org/abs/2506.15461

---

> > ### Author Rebuttal · Reviewer_rb5Y · 2026-04-03
> >
> > Dear Authors, thank you for the response with answers to my questions. I have follow up questions:
> >
> > 1. "In SPARe, each DP group is a fault-tolerance unit" - Does this mean that even if a single GPU fails within a DP group, the entire group is replaced by another group in the subsequent step? Could you please elaborate on this if I am missing something?
> >
> > 2. "TrainMover reduces the cost of each interruption via idle preheated GPUs (hot spares), whereas SPARe reduces the frequency of interruptions by masking failures at the DP layer." - SPARe replaces the failed DP unit with a replica and after this the communication group associated with gradient synchronization needs to adapt the new group. How does SPARe handle this? Seems like this process requires the same steps as resuming from an interruption.

---

> > > ### Author Response · Authors · 2026-04-05
> > >
> > > We thank the reviewer for the insightful acknowledgement, we faithfully answer to all the remaining concerns.
> > >
> > > # [Q1] Parallel group as a fault-tolerance unit
> > >
> > > **[Single GPU failure = group failure?] Yes,** our system setting is that even a single GPU failure breaks the entire group it belongs to. In fact, we are adopting that abstraction from Meta’s Llama 4 training system, **FT-HSDP [1],** which empirically validates this abstraction with their 98k GPUs cluster. Meta explains that the inner parallelism topology inside each group is very tightly coupled with globally coordinated intra-group computation and communication, hence in most cases **any node failure enforces the loss of the entire group [1]:** a tremendous challenge contemporary ML systems are facing that **SPARe** aims to overcome by **replicating data shards**.
> > >
> > > **[Does the failed group get replaced by another group?] No,** **SPARe** does **NOT** replace failed group with another group. When a group fails, **SPARe** continues the job without interruption using **only the surviving groups**. The redundancy in SPARe is not in the hardware units (groups), but in the **workloads (data shards)** assigned across them:  data shards are replicated and judiciously stacked across groups so that, even after multiple group losses, gradient synchronization can still proceed **without replacing the failed groups** as long as every shard type remains collectible **among the survivors**. Only when some shard type is completely wiped out, namely, when all groups hosting that shard type have failed under accumulated failures, does **SPARe** fall back to global restart and recovers all the failed groups. This is precisely a **key novelty** of **SPARe**: masking failures by replicating and replacing hardware indeed improves availability substantially, but is often either prohibitively expensive or requires significant systems engineering. **SPARe** instead converts the **hardware replication** into **data shard replication**, thereby achieving a similar availability gain with only 2~2.8x computation overhead.
> > >
> > > # [Q2] Difference with TrainMover
> > >
> > > As explained in **[Q1]**, **SPARe** does **NOT** replace a failed group with another group. Instead, when a group fails, **SPARe** simply **removes it** from the active job. The key point is that **SPARe** does **not** restore the hardware itself, but it restores **collectibility of all partial gradients** by stacking multiple **data shard replicas** across the groups as shown in **Fig 3 (a)** of manuscript. The surviving groups that host the same shard replicas as the failed group **execute additional shard stacks** to supply the missing partial gradients as shown in **Fig 3 (b-d)** of manuscript. Therefore, unlike **TrainMover**, which treats a failure as an interruption and then merges in a new group together with state transfer, **SPARe** keeps the active job running through failures and has the surviving groups substitute for the failed group **by computing extra shard stacks**. The role of adaptive reordering (**Sec 3.2**) is precisely to minimize how many extra stacks are needed so that all shard types (in other words, partial gradient types) remain collectible at the next gradient synchronization step. As shown by our theory in **Sec 4** and simulations in **Sec 5**, this average overhead remans only about 2~2.8 stacks over the course of training, even at high redundancy such as $r=20$ which would incur 20 additional stacks in traditional replication.
> > >
> > > We are deeply grateful to have this opportunity to clarify on our **SPARe** algorithm. **SPARe** substitutes the failed groups by having the **surviving groups take over the failed group’s job** by **executing additional stack** of data shard replicas. Data shards are  loaded and stored at each group **before** the job starts, hence system can keep running amidst failures without any need of data / state transfer. We hope we have adequately resolved the reviewer’s concern.
> > >
> > > [1] FT-HSDP: https://arxiv.org/abs/2602.00277

---

### Decision · Program_Chairs · 2026-04-30

**Decision:**

Accept (regular)

**Comment:**

After reading the reviews, rebuttal, and discussion, my recommendation is weak accept. The paper addresses an important systems problem for extreme-scale training and proposes a novel failure-masking approach based on redundant shard placement and adaptive reordering. The idea is interesting, and the accompanying theoretical analysis is a meaningful contribution.

The main remaining concerns are about practical realism. In particular, it remains unclear whether the failure model and simulation assumptions are sufficiently representative of real large-scale systems, especially under correlated failures and more complex bottlenecks. The abstraction at the data-parallel layer also leaves the inner topology and utilization implications somewhat underspecified, particularly for settings with high model-parallel degrees. In addition, the empirical comparison would be stronger with better baselines and a clearer utilization-oriented view of post-failure efficiency.

That said, I view these as limitations of scope and empirical completeness rather than fatal flaws in the core contribution. As a first step toward DP-layer failure masking at extreme scale, the paper is novel, technically interesting, and likely to stimulate useful follow-up work. For that reason, I land on weak accept.